# MMWorld: Towards Multi-discipline Multi-faceted World Model Evaluation in Videos

Xuehai He[1]    Weixi Feng[*2]    Kaizhi Zheng[*1]    Yujie Lu[*2]    Wanrong Zhu[*2]    Jiachen Li[*2]
Yue Fan[*1]    Jianfeng Wang[3]    Linjie Li[3]    Zhengyuan Yang[3]    Kevin Lin[3]
William Yang Wang[2]    Lijuan Wang[3]    Xin Eric Wang[1]
[1]UC Santa Cruz    [2]UC Santa Barbara    [3]Microsoft
{xhe89,xwang366}@ucsc.edu

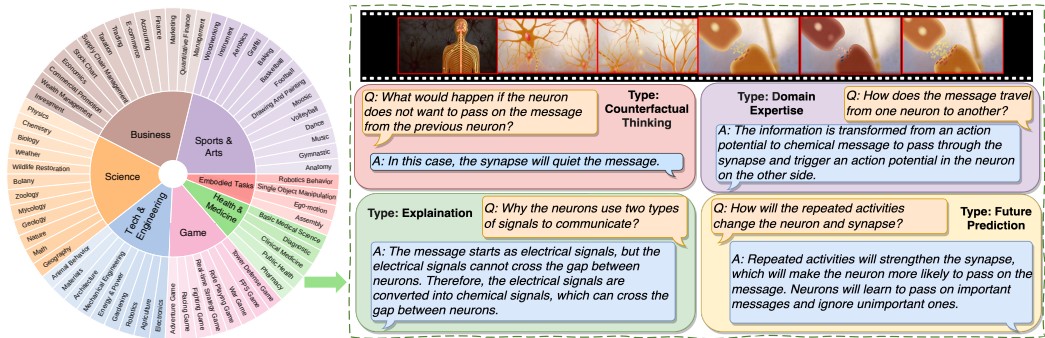

Figure 1: MMWorld covers seven broad disciplines and 69 subdisciplines, focusing on the evaluation of multi-faceted reasoning beyond perception (e.g., explanation, counterfactual thinking, future prediction, domain expertise). On the right is a video sample from the Health & Medicine discipline.

## Abstract

Multimodal Language Language Models (MLLMs) demonstrate the emerging abilities of "world models"—interpreting and reasoning about complex real-world dynamics. To assess these abilities, we posit videos are the ideal medium, as they encapsulate rich representations of real-world dynamics and causalities. To this end, we introduce MMWorld, a new benchmark for multi-discipline, multi-faceted multimodal video understanding. MMWorld distinguishes itself from previous video understanding benchmarks with two unique advantages: (1) **multi-discipline**, covering various disciplines that often require domain expertise for comprehensive understanding; (2) **multi-faceted reasoning**, including explanation, counterfactual thinking, future prediction, etc. MMWorld consists of a human-annotated dataset to evaluate MLLMs with questions about the whole videos and a synthetic dataset to analyze MLLMs within a single modality of perception. Together, MMWorld encompasses 1,910 videos across seven broad disciplines and 69 subdisciplines, complete with 6,627 question-answer pairs and associated captions. The evaluation includes 2 proprietary and 10 open-source MLLMs, which struggle on MMWorld (e.g., GPT-4V performs the best with only 52.3% accuracy), showing large room for improvement. Further ablation studies reveal other interesting findings such as models' different skill sets from humans. We hope MMWorld can serve as an essential step towards world model evaluation in videos.

---

*Equal Contribution

Submitted to the 38th Conference on Neural Information Processing Systems (NeurIPS 2024) Track on Datasets and Benchmarks. Do not distribute.

# 1   Introduction

Foundation models, such as Large Language Models (LLMs) [49; 59; 26; 2] and Multimodal LLMs (MLLMs) [51; 58; 36; 33; 45; 10], have demonstrated remarkable abilities in text and image domains, igniting debates about their potential pathways to Artificial General Intelligence (AGI). This raises a critical question: how well do these models understand the dynamics of the real world? Are they equipped with an inherent World Model [28; 11; 21; 65] that can understand and reason about the underlying principles and causalities of the dynamic, multimodal world?

Videos, with their rich, dynamic portrayal of the real world, are ideally suited for evaluating the "world modeling" capabilities of MLLMs. Existing video understanding benchmarks [34; 47; 53; 34], however, fall short in two key perspectives for such evaluations. First, as LeCun et al. [28] discussed, the world model should be able to *(1) estimate missing information about the state of the world not provided by perception, and (2) predict plausible future states of the world*. Evaluation of such capabilities requires **multi-faceted reasoning** beyond perception level, including explaining the video dynamics, counterfactual thinking of alternative consequences, and predicting future activities within videos. Moreover, the **multi-discipline** nature of the multimodal world necessitates a grasp of diverse fundamental principles—ranging from physics and chemistry to engineering and business. Hence, domain expertise across a variety of disciplines is imperative for a thorough evaluation of a model's world understanding towards AGI [46; 73].

Therefore, we introduce MMWorld, a multi-discipline multi-faceted multimodal video understanding benchmark for a comprehensive evaluation of MLLMs[2]. MMWorld encompasses a wide range of disciplines and presents multi-faceted reasoning challenges that demand a combination of visual, auditory, and temporal understanding. It consists of 1,910 videos that span seven common disciplines, including *Art & Sports*, *Business*, *Science*, *Health & Medicine*, *Embodied Tasks*, *Tech & Engineering*, and *Games*, and 69 subdisciplines (see Figure 1) such as Robotics, Chemistry, Trading, and Agriculture, thereby fulfilling the objective of breadth in discipline coverage. The dataset includes a total of 1,559 question-answer pairs and captions annotated and reviewed by humans. Meanwhile, for multi-faceted reasoning, MMWorld mainly contains seven kinds of questions focusing on *explanation* (explaining the phenomenon in videos), *counterfactual thinking* (answering what-if questions), *future prediction* (predicting future events), *domain expertise* (answering domain-specific inquiries), *temporal understanding* (reasoning about temporal information), and etc. A video example with these four questions from the Health & Medicine discipline is depicted in Figure 1. MMWorld comprises two datasets: a human-annotated dataset for evaluating MLLMs on the whole video and a synthetic dataset designed to analyze MLLMs' perception within single visual or audio modalities. We evaluate 12 MLLMs that can handle videos or image sequences on MMWorld, including both open-source (e.g., Video-LLaVA-7B [36]) and proprietary models (GPT-4V [51] and Gemini [58]).

We summarized the contributions and key findings as follows:

- We introduce MMWorld, a new benchmark designed to rigorously evaluate the capabilities of Multimodal Large Language Models (MLLMs) in world modeling through the realm of video understanding. MMWorld spans a broad spectrum of disciplines, featuring a rich array of question types for multi-faceted reasoning.

- In addition to the human-annotated dataset, we develop an automatic data collection pipeline, streamlining video content selection and question-answer generation, and construct a well-controlled synthetic dataset to analyze MLLMs within single visual or audio modalities.

- We observe that existing MLLMs still face substantial challenges posed by MMWorld. Even the best performer, GPT-4V, can only achieve a 52.30% overall accuracy, and four MLLMs particularly trained on videos perform worse than random chance.

- Although there is stll a clear gap between open-source and proprietary models, the best open-source model Video-LLaVA-7B outperforms GPT-4V and Gemini on Embodied Tasks by a large margin

---

[2]Note that MMWorld is not a sufficient testbed for world model evaluation, but we believe overcoming the unique challenges presented in MMWorld is essential and necessary towards comprehensive world modeling.

Table 1: Comparison between MMWorld and previous benchmarks for real-world video understanding on a variety of criteria. Multi-faced include Explanation (`Explain.`), Counterfactual Thinking (`Count.`), Future Prediction (`Future.`) and Domain Expertise (`Domain.`) MMWorld is the first multi-discipline and multitask video understanding benchmark that covers wider reasoning questions, and also included first-party data annotations.

| Benchmarks | Multi-Discipline | Multi-Task | Multi-Faceted Reasoning | | | | First-Party Annotation |
|---|---|---|---|---|---|---|---|
| | | | Explain. | Count. | Future. | Domain. | |
| MovieQA [57], TVQA [29] | | | ✓ | | | | ✓ |
| ActivityNet-QA [71] | | | | | | | ✓ |
| MSVD-QA [66], MSRVTT-QA [67] | | | | | | | ✓ |
| Sports-QA [31] | | | | ✓ | | ✓ | ✓ |
| VaTeX [61] | | ✓ | | | | | ✓ |
| VALUE [35] | | ✓ | | | | | |
| Video-Bench [48] | | ✓ | | | ✓ | ✓ | |
| MVBench [34] | | ✓ | | ✓ | ✓ | | |
| Perception Test [53] | | ✓ | ✓ | ✓ | ✓ | | |
| MMWorld (Ours) | ✓ | ✓ | ✓ | ✓ | ✓ | ✓ | ✓ |

and performs similarly on Art & Sports, where spatiotemporal dynamics play a more crucial role in video understanding. This is further validated with its leading results on the Temporal Understanding question type.

- In our study comparing MLLMs with average humans (non-experts), we notice some correlation between question difficulties as perceived by humans and MLLMs. However, MLLMs present different skill sets than humans in that they can answer reasonable amount of difficult questions that humans completely fail but also struggle at easy questions that humans excel at. This indicates different perception, cognition, and reasoning abilities between MLLMs and humans.

## 2 Related Work

### 2.1 Multimodal Large Language Models (MLLMs)

**Emerging MLLMs** With recent breakthroughs [50; 18; 59; 12; 60; 4] in Large Language Models (LLMs), several counterparts in the vision-and-language domain have been proposed [14; 41; 40; 30; 78; 77; 5], and recently released GPT-4V [51], followed by Gemini Vision family [58]. Many MLLMs have expanded their capabilities beyond handling only text and image inputs. VideoChat [33] leverages the QFormer [32] to map visual representations to LLM [12], and performs a multi-stage training pipeline. Otter [30] proposes to conduct instruction finetuning based on Openflamingo [3]. PandaGPT [56] employs the ImageBind [23] as the backbone and finetunes it. mPLUG-Owl [68] introduces an abstractor module to perform visual and language alignment. VideoLLaMA [75] introduces a frame embedding layer and also leverages ImageBind to inject temporal and audio information into the LLM backend. Chat-UniVi [27] uses clustering to do feature fusion. Observing their emerging abilities in multimodal video understanding, we propose MMWorld to evaluate these models' skills in understanding the dynamics of the real world.

**Benchmarking MLLMs** To evaluate MLLMs, there is a flourishing of analysis [38; 76; 43; 15; 13; 20; 70; 16] and the establishment of innovative benchmarks such as VisIB-Bench [8] which evaluates models with real-world instruction-following ability given image inputs, MMMU [73] designed to access models on college-level image-question pairs that span among different disciplines, and VIM [44] which challenges the model's visual instruction following capability. However, these recent analyses and benchmarks only cover the image input, which hinders the evaluation of MLLM's performance as a world model. Recently, video benchmarks such as Perception Test [53] is proposed to focus on perception and skills like memory and abstraction. However, it uses scenarios with a few objects manipulated by a person, which limits the variety of contexts. MVBench [34] centers on temporal understanding, while MMWorld not only includes temporal reasoning but also evaluates other multi-faceted reasoning abilities.

## 2.2 Video Understanding Benchmarks

Previous video benchmarks, as shown in Table 1, focus on video understanding tasks, including activity-focused on web videos [72], description-based question answering [74], video completion [17], and video infilling [24]. Recently, Video-Bench [47] introduces a benchmark by collecting videos and annotations from multiple existing datasets. LWM [39] collects a large video and language dataset from public books and video datasets and trains a world model that is capable of processing more than millions of tokens. However, modeling millions of tokens is extremely difficult due to high memory cost, computational complexity, and lack of suitable datasets. Mementos [62] builds a benchmark for MLLM reasoning for input image sequences. STAR [64] builds a benchmark for situated reasoning in real-world videos. CLEVER [69] builds a benchmark containing videos focusing on objects with simple visual appearance. Our contribution, in contrast, presents a new video understanding benchmark designed to evaluate models on several pivotal components crucial for a comprehensive world model. These components encompass interdisciplinary coverage, task diversity, and multifaceted reasoning capabilities—including future prediction, counterfactual thinking, and more—underpinned by original human annotations and integrated domain knowledge.

## 3 The MMWorld Benchmark

The MMWorld benchmark is built on three key design principles: multi-discipline coverage and multi-faceted reasoning. It spans various disciplines that require domain expertise and incorporates diverse reasoning skills such as explanation, counterfactual thinking, and future prediction. The benchmark consists of two parts: a human-annotated dataset and a synthetic dataset. The human-annotated dataset serves as the main test bed to evaluate MLLMs from multiple perspectives. The synthetic dataset contains two subsets, focusing on evaluating MLLMs' perception behavior from both visual signals and audio inputs, respectively.

### 3.1 Manual Data Collection

We collect videos from YouTube with the Creative Licence in seven disciplines: Art & Sports (18.5%), Business (12.0%), Science (20.4%), Health & Medicine (12.0%), Embodied Tasks (12.0%%), Tech & Engineering (12.9%), and Game (12.2%). For Art & Sports, 29 videos are collected from the SportsQA dataset [31]. And for Embodied Tasks, 24 videos are sourced from IKEA Assembly [7], RT-1 [9], and Ego4D [19] datasets to increase video diversity.

Our manual benchmark collection takes two stages. In the first stage, we conduct a detailed examination of each of the seven primary disciplines to identify a comprehensive range of subdisciplines for inclusion in our benchmark. Our selection of videos is driven by two key principles:

The **first principle**, **multi-discipline** coverage, emphasizes the requirement for domain knowledge—selecting videos that inherently demand an understanding of specialized content across various disciplines. The **second principle**, **multi-faceted** annotation, involves collecting videos that enable the creation of question-answer pairs from multiple perspectives to evaluate world model properties comprehensively. The **third principle**, **temporal information**, prioritizes the inclusion of videos that provide meaningful content over time, as understanding temporal information is crucial for grasping world dynamics. This allows models to engage in temporal reasoning. Therefore, answering questions in our dataset requires implicit temporal reasoning, e.g., the model needs to understand temporal information to explain "why does the robot need to do the step shown in the video". We also design a "temporal understanding" question type to explicitly test models' ability to reason about temporal information (examples can be found in Section F in the Appendix).

During the second stage, our team embark on the task of question annotation. We craft questions that primarily test seven aspects of multimodal video understanding also from the perspective of **multi-faceted reasoning**: 1) Explanation: Questions ask the model to elucidate the underlying logic or purpose within the video; 2) Counterfactual Thinking: Tests the model's ability to hypothesize and consider alternative outcomes; 3) Future Prediction: Aims to predict future events based on the current

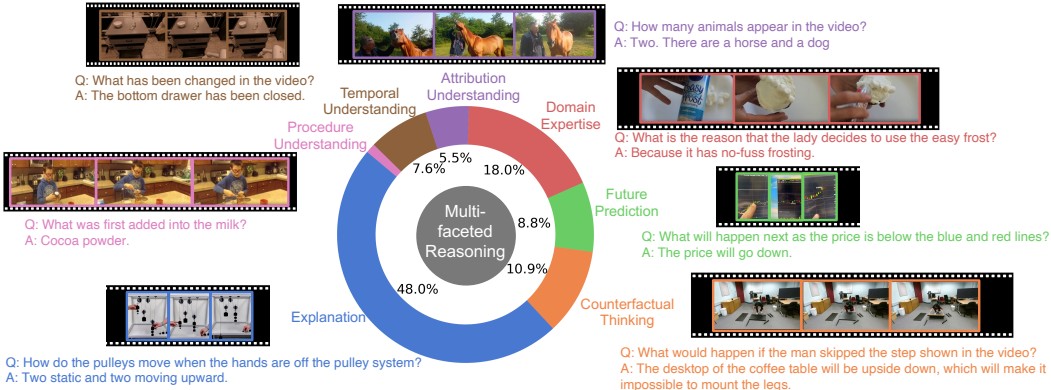

Figure 2: The questions in MMWorld primarily evaluate seven understanding and reasoning abilities of models to provide correct answers.

scenario, challenging the model's foresight; 4) Domain Expertise: Evaluates the model's depth of knowledge in specific fields, such as how to assemble a coffee table; 5) Temporal Understanding: Assesses the model's capability to reason about temporal sequences and dynamics; 6) Attribution Understanding: These questions focus on identifying cause-and-effect relationships within the video, including tasks like counting; 7) Procedure Understanding: Tests the model's ability to comprehend and explain procedural tasks shown in the video. The detailed distribution and examples are shown in Figure 2.

## 3.2 Automated Data Collection

Table 2: Key Statistics of the MMWorld Benchmark. The main subset is the human-annotated subset. Synthetic Subset I contains generated QA pairs focused exclusively on the audio content, while Synthetic Subset II contains QA pairs focused exclusively on the visual content of the video.

| Statistics | Main Subset | Synthetic I | Synthetic II |
|---|---|---|---|
| #Discipline/#Subdiscipline | 7/61 | 7/51 | 7/54 |
| #<Video-QA> | <417-1,559> | <746-2,969> | <747-2,099> |
| Avg Video Lengths (s) | 102.3 | 103.4 | 115.8 |
| Avg #Questions per Video | 4.05 | 3.98 | 2.81 |
| Avg #Options | 3.90 | 4.00 | 4.00 |
| Avg Question Length | 11.39 | 15.12 | 17.56 |
| Avg Option Length | 7.27 | 6.01 | 5.19 |
| Avg Answer Length | 6.42 | 6.71 | 5.67 |
| Avg Caption Length | 27.00 | 71.87 | 82.33 |

Understanding real-world dynamics requires models to process both audio and visual modalities. To evaluate MLLMs' perception abilities in these modalities, we designed an automated data collection pipeline. This pipeline collects targeted videos and generates QA pairs based on either audio or visual information, ensuring the model's capabilities are assessed independently for each modality. By using a single modality's information for generation, our pipeline ensures that the synthetic data remains unbiased regarding input modality.

The synthetic data generation pipeline is illustrated in Figure 3. We employ a systematic approach to gather videos with Creative Commons licenses from YouTube and the extensive YouTube-8M dataset [1]. This method ensures a diverse and comprehensive collection of video data, which is important for the robust evaluation of multimodal video understanding models.

**Video Collection and Processing** We start with the video *Query Generator*. We start with the same seven disciplines as the manually collected dataset. For each discipline, a set of subdisciplines is defined to encapsulate a wide spectrum of topics, ensuring a diverse and comprehensive dataset. Once the queries are generated, the *Video Mapping and Filtering* step is initiated. We perform mapping of videos to YouTube-8M and online videos, constrained by a strict time limit of two minutes per query, keeping only the most pertinent videos that satisfy the predefined criteria. Simultaneously, the works in conjunction with the video transcripts to extract key terms and concepts. This iterative process refines the search parameters and enhances the semantic richness of the dataset by identifying and encoding the salient themes present in the videos. The *Video Summarization* module utilizes Query-

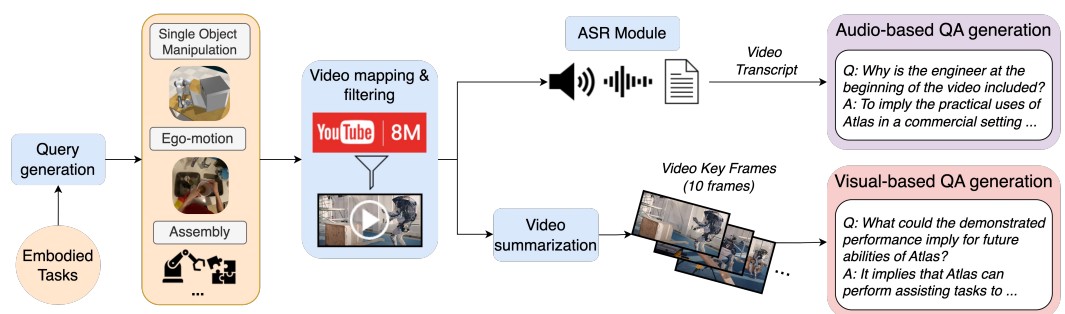

Figure 3: Schematic diagram of the synthetic data generation pipeline in MMWorld. It starts with generating subdiscipline-specific queries, followed by video retrieval from YouTube-8M [1] and YouTube. Keyframes are extracted for visual-based QA generation, and videos are transcribed using an ASR module for audio-based QA generation.

focused video summarization techniques based on Katna[3] and UniVTG [37]. This module selects ten representative frames from each video, distilling the essence of the content while preserving the narrative context. This summarization facilitates efficient storage and quicker processing times, which are crucial for large-scale analysis.

**QA Generation** The final stage in our pipeline is the *QA / Caption Generation* module, where we leverage the capabilities of GPT-4V to generate accurate and contextually relevant questions and answers, as well as captions, based on the video frames and transcripts. This step not only provides rich annotations for each video but also equips the dataset with a multimodal dimension that supports various downstream tasks such as video QA, captioning, and more.

**Quality of the Synthetic Dataset** Human evaluators were engaged to ascertain the reasonableness of automatically generated questions and answers, ensuring that the synthetic dataset maintains a high standard of quality and relevance. The findings from this human evaluation phase are detailed in Section 3 of the Appendix, offering insights into the dataset's efficacy and the realism of its constructed queries and responses.

Finally, the statistics of automated curated data, which is used for the ablation study, are shown in Table 2. The taxonomy of our dataset is shown in Figure 1. We note that only a portion of the subdisciplines are shown due to space concerns. Please refer to the Appendix for full information.

## 4 Experiments

### 4.1 Experimental Settings

In our study, we compare MLLM's performance on the MMWorld benchmark, including GPT-4V [51], Gemini Pro [58], Video-Chat [33], Video-LLaMA [75], ChatUnivi [27], mPLUG-Owl [68], Otter [30], ImageBind-LLM [23], PandaGPT [56], LWM [39], and X-Instruct-BLIP [52]. For both Gemini Pro and GPT-4V, we adhere to the default settings provided by their official APIs. They both take ten image frames extracted from the video content as the input. The Gemini Pro is set to process visual input and configured with safety settings to filter a range of harmful content. The configuration thresholds are set to 'BLOCK_NONE'. For PandaGPT, we set 'top_p' to 0.7 and 'temperature' to 0.5. For VideoChat, we set 'max_frames' to 100. For X-Instruct-BLIP, the model is implemented using four image frames. We use GPT-4-32K as the judge for judging whether the model answer is correct when it can not mapped to the option letter using the rule-based method. For others, we all use the default setting. All inferences are run on a NVIDIA A6000 workstation. The detailed implementation is given in the Appendix.

---

[3]`https://github.com/keplerlab/katna`

Table 3: MLLM accuracy across diverse disciplines (averaging over three runs). GPT-4V and Gemini Pro lead at most disciplines and achieve the best overall accuracy. The best open-source model Video-LLaVA-7B outperforms them on Embodied Tasks and perform similarly on Art & Sports.

| Model | Art& Sports | Business | Science | Health& Medicine | Embodied Tasks | Tech& Engineering | Game | Average |
|---|---|---|---|---|---|---|---|---|
| Random Choice | 25.03 | 25.09 | 26.44 | 25.00 | 26.48 | 30.92 | 25.23 | 26.31 |
| *Proprietary MLLMs* | | | | | | | | |
| GPT-4V [51] | 36.17 ±0.58 | **81.59** ±1.74 | **66.52** ±1.86 | 73.61 ±0.49 | 55.48 ±2.70 | 61.35 ±1.00 | **73.49** ±1.97 | **52.30** ±0.49 |
| Gemini Pro [58] | 37.12 ±2.68 | 76.69 ±2.16 | 62.81 ±1.83 | **76.74** ±1.30 | 43.59 ±0.33 | **69.86** ±2.01 | 66.27 ±2.60 | 51.02 ±1.35 |
| *Open-source MLLMs* | | | | | | | | |
| Video-LLaVA-7B [36] | 35.91 ±0.96 | 51.28 ±0.87 | 56.30 ±0.76 | 32.64 ±0.49 | **63.17** ±1.44 | 58.16 ±1.00 | 49.00 ±3.16 | 44.60 ±0.58 |
| Video-Chat-7B [33] | **39.53** ±0.06 | 51.05 ±0.00 | 30.81 ±0.21 | 46.18 ±0.49 | 40.56 ±0.57 | 39.36 ±0.00 | 44.98 ±0.57 | 40.11 ±0.06 |
| ChatUnivi-7B [27] | 24.47 ±0.49 | 60.84 ±1.51 | 52.00 ±0.73 | 61.11 ±1.96 | 46.15 ±2.06 | 56.74 ±1.33 | 52.61 ±2.84 | 39.47 ±0.42 |
| mPLUG-Owl-7B [68] | 29.16 ±1.62 | 64.10 ±1.84 | 47.41 ±3.29 | 60.07 ±1.30 | 23.78 ±3.47 | 41.84 ±5.09 | 62.25 ±3.16 | 38.94 ±1.52 |
| PandaGPT-7B [56] | 25.33 ±0.54 | 42.66 ±3.02 | 39.41 ±2.67 | 38.54 ±3.07 | 35.43 ±0.87 | 41.84 ±2.79 | 40.16 ±4.65 | 32.48 ±0.45 |
| ImageBind-LLM-7B [23] | 24.82 ±0.16 | 42.66 ±0.99 | 32.15 ±1.11 | 30.21 ±1.47 | 46.85 ±1.14 | 41.49 ±1.50 | 41.37 ±0.57 | 31.75 ±0.14 |
| X-Instruct-BLIP-7B [52] | 21.08 ±0.27 | 15.85 ±0.87 | 22.52 ±1.11 | 28.47 ±0.49 | 18.41 ±1.44 | 22.34 ±0.87 | 26.10 ±0.57 | 21.36 ±0.18 |
| LWM-1M-JAX [39] | 12.04 ±0.53 | 17.48 ±0.57 | 15.41 ±0.91 | 20.49 ±0.98 | 25.87 ±1.98 | 21.99 ±2.19 | 11.65 ±3.01 | 15.39 ±0.32 |
| Otter-7B [30] | 17.12 ±1.17 | 18.65 ±0.87 | 9.33 ±0.36 | 6.94 ±0.98 | 13.29 ±1.51 | 15.96 ±1.74 | 15.26 ±0.57 | 14.99 ±0.77 |
| Video-LLaMA-2-13B [75] | 6.15 ±0.44 | 21.21 ±0.66 | 22.22 ±1.45 | 31.25 ±1.70 | 15.38 ±1.14 | 19.15 ±1.74 | 24.90 ±5.93 | 14.03 ±0.29 |

## 4.2 Evaluation

Our dataset includes multiple-choice questions and captions corresponding to each video, enabling tasks such as video question answering and video captioning. We focus on video question answering by evaluating a model's performance based on its accuracy in selecting the correct answer from the provided options. One challenge lies in reliably parsing the model's response to map it to one of the predefined choices. To address this, we employ two mapping strategies. We employ two mapping strategies. The first method employs automated scripts to parse the models' predictions and compare the parsed results with the ground truth, similar to the approach used in [73]. The second method involves models freely generating answers, which are then evaluated by GPT-4. Given the question, correct answer, and model's prediction, GPT-4 returns a True or False judgment. This approach is based on recent works in model evaluation [45; 25; 22; 42]. We validated this method with human evaluators, showing an error rate of 4.76% across 189 examples, confirming the effectiveness of GPT-4 as an evaluator. Detailed results for human evaluation and for these two different strategies are provided in Appendix B. In the main paper, all results are evaluated using the second approach.

## 4.3 Main Evaluation Results

We show in Table 3 the main evaluation results of different MLLMs. Among these, GPT-4V emerges as the top performer, closely followed by Gemini Pro. Video-LLaVA also demonstrates strong results, primarily due to the extensive training data which consists of 558K LAION-CCSBU image-text pairs and 702K video-text pairs from WebVid [6]. For instruction tuning, datasets were gathered from two sources: a 665K image-text instruction dataset from LLaVA v1.5 and a 100K video-text instruction dataset from Video-ChatGPT [45]. This superior performance may also be attributed to Video-LLaVA's adoption of CLIP ViT-L/14 trained in LanguageBind [36] as its vision model and the inclusion of a large volume of image-video-text pairings within the training data. On the other hand, models like Otter and LWM perform poorly across most disciplines, possibly due to their weaker backbone and architecture used. Otter uses the LLaMA-7B language encoder and a CLIP ViT-L/14 vision encoder, both of which are frozen, with only the Perceiver resampler module fine-tuned, which may contribute to its lower performance. Additionally, some MLLMs perform even worse than random, highlighting the challenging nature of MMWorld.

## 4.4 Study on Multi-faceted Reasoning on MMWorld

Figure 4 illustrates the multi-faceted reasoning performance for each MLLM. GPT-4V emerges as the strongest model across Future Prediction, Domain Expertise, and Attribution Understanding.

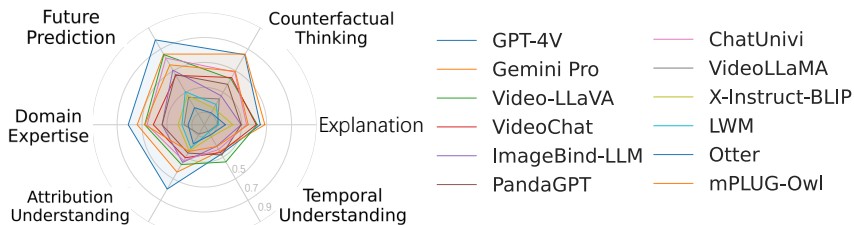

Figure 4: Results of different MLLMs on multi-faceted reasoning. The detailed performance numbers can be found in the Appendix.

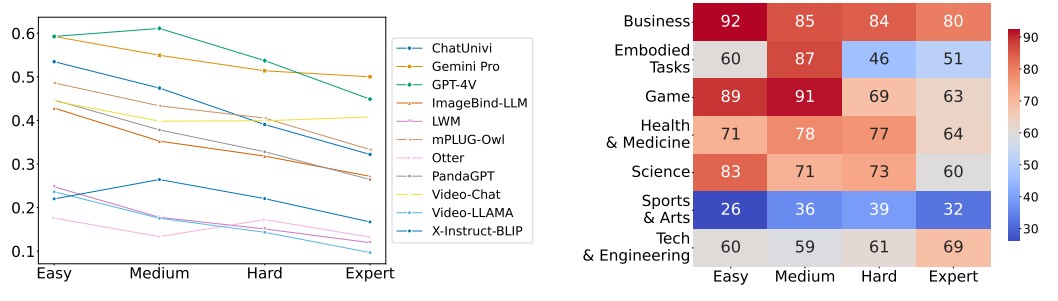

(a) Accuracy of MLLMs at difficulty levels.  (b) GPT-4V results by disciplines at difficulty levels.

Figure 5: Model performance at different difficulty levels for average humans. Average human difficulty levels are defined by 3 turkers' performance per question: Easy (3/3 correct answers), medium (2/3 correct), hard (1/3 correct), and expert (0/3 correct).

Closed-source models like GPT-4V and Gemini Pro perform similarly on counterfactual thinking and outperform all others. However, for temporal understanding, Video-LLaVA performs the best. This may be due to its extensive training on large amounts of video-language data, which enhances its spatio-temporal reasoning abilities. This can be also observed in its high scores on the Art & Sports and Embodied Tasks, which involve dense spatio-temporal information, as shown in Table 3. Video-LLaVA's performance is comparable to GPT-4V and Gemini on explanation tasks, likely because of its two-stage training process and exposure to a large amount of instruction-tuning data in the second stage, which includes similar instructions.

## 4.5 Study on MLLM Performance at Different Difficulty Levels for Average Humans

Figure 5a indicate some correlation between the difficulty levels as perceived by humans and the performance of MLLMs. MLLMs generally follow a trend where accuracy decreases as the difficulty level increases, which aligns with human performance patterns. However, the correlation is not perfect, suggesting that while models and humans share some common ground in understanding question difficulty, there are also notable differences in their capabilities. The data reveals that MLLMs exhibit different skill sets compared to humans. As highlighted in Figure 5b, models like GPT-4V can correctly answer expert-level questions that humans often get wrong, particularly in disciplines such as Business and Health & Medicine, where humans often struggle, yet they sometimes falter on easier questions, likely due to the lack of contextual understanding. Notably, discrepancies in disciplines like Art & Sports and Tech & Engineering highlight areas where MLLMs' performance does not align with human results, suggesting different perception, cognition, and reasoning abilities in handling abstract concepts. These differences suggest that MLLMs can complement human capabilities, offering potential for enhanced task performance by combining the data-driven insights of models with human intuition and contextual knowledge.

Table 4: Performance on Synthetic Subsets I (Audio) and II (Visual). Synthetic Subset I contains QAs based solely on the audio content, while Synthetic Subset II focuses exclusively on the visual content of the video. We evaluated four MLLMs that can process both audio and visual inputs along with Gemini Pro (for the audio setting, only providing the question).

| Model | Art&Sports | | Business | | Science | | Health&Medicine | | Embodied Tasks | | Tech&Engineering | | Game | | Average | |
|---|---|---|---|---|---|---|---|---|---|---|---|---|---|---|---|---|
| | Audio | Visual | Audio | Visual | Audio | Visual | Audio | Visual | Audio | Visual | Audio | Visual | Audio | Visual | Audio | Visual |
| Random Choice | 31.59 | 30.14 | 31.18 | 26.58 | 36.98 | 32.89 | 38.74 | 32.64 | 32.81 | 31.25 | 27.23 | 32.60 | 32.01 | 30.78 | 32.44 | 30.91 |
| Video-Chat [33] | **33.98** | 32.48 | **46.47** | 41.46 | **41.86** | 39.15 | **45.95** | 36.81 | 32.81 | 46.88 | **37.48** | 35.91 | **32.98** | 46.70 | **38.82** | 39.07 |
| ChatUnivi [27] | 30.03 | 43.22 | 30.19 | 52.85 | 38.75 | 54.59 | 34.76 | 50.69 | 20.14 | 40.63 | 24.17 | 46.41 | 29.98 | 45.44 | 31.82 | 48.44 |
| Video-LLaMA [75] | 30.15 | 30.23 | 36.18 | 33.17 | 31.33 | 31.34 | 30.90 | 32.78 | **33.13** | 30.05 | 31.18 | 30.55 | 20.49 | 27.20 | 29.08 | 30.47 |
| Otter [30] | 14.22 | 16.82 | 16.77 | 14.24 | 16.12 | 17.00 | 19.82 | 13.19 | 10.94 | 12.50 | 15.63 | 12.43 | 6.65 | 10.44 | 12.83 | 13.41 |
| Gemini Pro [58] | 20.88 | **61.38** | 29.43 | **77.35** | 30.62 | **74.26** | 30.14 | **81.53** | 22.57 | **70.31** | 18.83 | **66.22** | 29.96 | **65.01** | 24.45 | **69.97** |

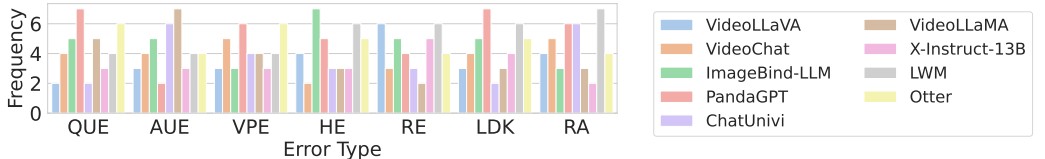

Figure 6: The frequency of different error types across various MLLMs. For each error type, 10 examples were evaluated. Error types are abbreviated as follows: QUE (Question Understanding Error), AUE (Audio Understanding Error), VPE (Visual Perception Error), HE (Hallucination Error), RE (Reasoning Error), LDK (Lack of Domain Knowledge), and RA (Reject to Answer).

### 4.6 Study on Modality of Perception

We conduct ablations to evaluate MLLMs ability to perceiving the world on the synthetic dataset of MMWorld. With our synthetic dataset, we considered scenarios where only one modality—either audio or visual—is available. Table 4 shows the results which evaluates the model's ability to interpret spoken language, background noises, and other audio elements without the aid of visual context and the model's perception ability to operate without any audio input. For the visual perception test, Gemini Pro performed the best, demonstrating its strong ability to process visual information. Interestingly, Video-Chat exhibited better audio perception than ChatUnivi, despite its poorer visual perception. This may be attributed to its use of the Whisper [54] speech recognition model. It also explains that in Table 3, Video-Chat outperforms ChatUnivi in the Art & Sports discipline, which requires a greater understanding of music, voice, and background audio. However, in other disciplines such as Science and Health & Medicine, Video-Chat's performance is significantly poorer.

### 4.7 Error Analysis

To gain deeper insights into the limitations of MLLMs, we prompted the models to explain the reasoning behind their choices, particularly when errors occurred. Through this analysis, we identified common error patterns and summarized them into seven distinct categories. We conducted a simple test where the same questions that triggered errors in GPT-4V were also posed to other MLLMs. The frequencies of each type of error are presented in Figure 6, as annotated by human evaluators. Detailed qualitative examples of these errors and further analysis are provided in the Appendix.

## 5 Conclusion

Our MMWorld Benchmark represents a significant step forward in the quest for advanced multi-modal language models capable of understanding complex video content. By presenting a diverse array of videos across seven disciplines, accompanied by questions that challenge models to demonstrate explanation, counterfactual thinking, future prediction, and domain expertise, we have created a rigorous testing ground for the next generation of AI. While using LLMs for data generation can introduce hallucination issues, these challenges are manageable and are commonly addressed [63; 55]. Another potential risk is the misuse of MLLMs for surveillance or privacy invasion. The ability of models to understand video content and perform reasoning could be exploited to monitor individuals without their consent, leading to serious ethical and legal concerns regarding privacy.

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
