# MMWorld: Towards Multi-discipline Multi-faceted World Model Evaluation in Videos — Appendix

**Xuehai He**[1] **Weixi Feng**[*2] **Kaizhi Zheng**[*1] **Yujie Lu**[*2] **Wanrong Zhu**[*2] **Jiachen Li**[*2]
**Yue Fan**[*1] **Jianfeng Wang**[3] **Linjie Li**[3] **Zhengyuan Yang**[3] **Kevin Lin**[3]
**William Yang Wang**[2] **Lijuan Wang**[3] **Xin Eric Wang**[1]
[1]UC Santa Cruz    [2]UC Santa Barbara    [3]Microsoft
{xhe89,xwang366}@ucsc.edu

## A    Overview of the Appendix

We host the project website on `https://mmworld-bench.github.io/`. The benchmark and code implementations can be found at `https://github.com/eric-ai-lab/MMWorld`. The link to Croissant metadata record documenting the dataset/benchmark available for viewing and downloading is available at `https://github.com/eric-ai-lab/MMWorld/blob/main/data/croissanta_hf_data.json`. This Appendix is organized as follows:

- Section B contains additional experimental results;
- Section C contains the implementation details;
- Section D contains the settings and results from human evaluations;
- Section E contains the error analysis;
- Section F contains the data examples from MMWorld;
- Section G contains additional data statistics of MMWorld;
- Section H contains the datasheet of MMWorld;
- Section I contains the author statement, licence, and maintenance plan.

## B    Additional Results

### B.1    Results Across Different Seed for Each Model

In Table 1, we show detailed results using three different seeds for each evaluated models.

### B.2    Results from Amazon Turkers

Table 2 presents the evaluation results from three sets of Amazon Turkers across various disciplines. The results indicate that there is slightly variability in performance across different human evaluators.

### B.3    Results for the Two Different Evaluation Strategies

In Table 3, we give additional evaluation results for different MLLMs evaluated in this paper. For closed-source models, the evaluation pipeline is the one used in the main paper, which involves

---

*Equal Contribution

Submitted to the 38th Conference on Neural Information Processing Systems (NeurIPS 2024) Track on Datasets and Benchmarks. Do not distribute.

Table 1: Detailed results of model performance, measured as accuracy percentages across diverse disciplines for three runs. The random choice baseline involves shuffling candidate answers for each video question before consistently selecting answer 'a'. GPT-4V and Gemini Pro utilize 10 image frames extracted from the video content.

| Model | Art&Sports | Business | Science | Health&Medicine | Embodied Tasks | Tech&Engineering | Game | Average |
|---|---|---|---|---|---|---|---|---|
| GPT-4V-seed 1 [7] | 36.90 | 79.72 | 64.00 | 73.96 | 51.75 | 60.64 | 71.08 | 51.64 |
| GPT-4V-seed 2 [7] | 35.48 | 83.92 | 68.44 | 73.96 | 58.04 | 60.64 | 75.90 | 52.79 |
| GPT-4V-seed 3 [7] | 36.13 | 81.12 | 67.11 | 72.92 | 56.64 | 62.77 | 73.49 | 52.47 |
| Gemini Pro-seed 1 [10] | 40.90 | 79.72 | 60.44 | 78.12 | 43.36 | 71.28 | 65.06 | 52.92 |
| Gemini Pro-seed 2 [10] | 35.10 | 75.52 | 63.11 | 75.00 | 44.06 | 71.28 | 69.88 | 50.16 |
| Gemini Pro-seed 3 [10] | 35.35 | 74.83 | 64.89 | 77.08 | 43.36 | 67.02 | 63.86 | 49.97 |
| Video-LLaVA-seed 1 [5] | 34.58 | 51.05 | 57.33 | 32.29 | 61.54 | 57.45 | 50.60 | 43.94 |
| Video-LLaVA-seed 2 [5] | 36.77 | 52.45 | 56.00 | 32.29 | 65.03 | 57.45 | 51.81 | 45.35 |
| Video-LLaVA-seed 3 [5] | 36.39 | 50.35 | 55.56 | 33.33 | 62.94 | 59.57 | 44.58 | 44.52 |
| Video-Chat-seed 1 [4] | 39.48 | 51.05 | 30.67 | 46.88 | 39.86 | 39.36 | 44.58 | 40.03 |
| Video-Chat-seed 2 [4] | 39.48 | 51.05 | 30.67 | 45.83 | 41.26 | 39.36 | 45.78 | 40.15 |
| Video-Chat-seed 3 [4] | 39.61 | 51.05 | 31.11 | 45.83 | 40.56 | 39.36 | 44.58 | 40.15 |
| mPLUG-Owl-seed 1 [11] | 31.35 | 65.73 | 45.78 | 61.46 | 28.67 | 48.94 | 65.06 | 41.05 |
| mPLUG-Owl-seed 2 [11] | 28.65 | 65.03 | 44.44 | 58.33 | 21.68 | 37.23 | 57.83 | 37.52 |
| mPLUG-Owl-seed 3 [11] | 27.48 | 61.54 | 52.00 | 60.42 | 20.98 | 39.36 | 63.86 | 38.23 |
| ChatUnivi-seed 1 [2] | 24.13 | 60.14 | 52.00 | 62.50 | 48.95 | 56.38 | 56.63 | 39.77 |
| ChatUnivi-seed 2 [2] | 25.16 | 62.94 | 51.11 | 62.50 | 44.06 | 58.51 | 50.60 | 39.77 |
| ChatUnivi-seed 3 [2] | 24.13 | 59.44 | 52.89 | 58.33 | 45.45 | 55.32 | 50.60 | 38.87 |
| PandaGPT-seed 1 [9] | 26.06 | 44.06 | 38.22 | 41.67 | 35.66 | 39.36 | 42.17 | 32.97 |
| PandaGPT-seed 2 [9] | 24.77 | 45.45 | 36.89 | 34.38 | 34.27 | 40.43 | 44.58 | 31.88 |
| PandaGPT-seed 3 [9] | 25.16 | 38.46 | 43.11 | 39.58 | 36.36 | 45.74 | 33.73 | 32.58 |
| ImageBind-LLM-seed 1 [1] | 24.77 | 41.96 | 30.67 | 31.25 | 46.85 | 43.62 | 40.96 | 31.62 |
| ImageBind-LLM-seed 2 [1] | 25.03 | 41.96 | 32.44 | 31.25 | 45.45 | 40.43 | 40.96 | 31.69 |
| ImageBind-LLM-seed 3 [1] | 24.65 | 44.06 | 33.33 | 28.12 | 48.25 | 40.43 | 42.17 | 31.94 |
| X-Instruct-BLIP-seed 1 [8] | 21.42 | 14.69 | 22.22 | 29.17 | 16.78 | 21.28 | 26.51 | 21.23 |
| X-Instruct-BLIP-seed 2 [8] | 20.77 | 16.78 | 24.00 | 28.12 | 20.28 | 22.34 | 25.30 | 21.62 |
| X-Instruct-BLIP-seed 3 [8] | 21.03 | 16.08 | 21.33 | 28.12 | 18.18 | 23.40 | 26.51 | 21.23 |
| LWM-seed 1 [6] | 11.35 | 18.18 | 16.44 | 19.79 | 24.48 | 24.47 | 10.84 | 15.20 |
| LWM-seed 2 [6] | 12.13 | 17.48 | 15.56 | 19.79 | 24.48 | 22.34 | 8.43 | 15.14 |
| LWM-seed 3 [6] | 12.65 | 16.78 | 14.22 | 21.88 | 28.67 | 19.15 | 15.66 | 15.84 |
| Otter-seed 1 [3] | 18.45 | 19.58 | 8.89 | 8.33 | 14.69 | 15.96 | 14.46 | 15.84 |
| Otter-seed 2 [3] | 17.29 | 17.48 | 9.33 | 6.25 | 13.99 | 18.09 | 15.66 | 15.14 |
| Otter-seed 3 [3] | 15.61 | 18.88 | 9.78 | 6.25 | 11.19 | 13.83 | 15.66 | 13.98 |
| Video-LLaMA-seed 1 [12] | 5.55 | 21.68 | 24.00 | 29.17 | 15.38 | 21.28 | 18.07 | 13.66 |
| Video-LLaMA-seed 2 [12] | 6.58 | 20.28 | 20.44 | 31.25 | 13.99 | 17.02 | 32.53 | 14.05 |
| Video-LLaMA-seed 3 [12] | 6.32 | 21.68 | 22.22 | 33.33 | 16.78 | 19.15 | 24.10 | 14.37 |

Table 2: Performance of different set of turkers

| Model | Art&Sports | Business | Science | Health&Medicine | Embodied Tasks | Tech&Engineering | Game& | Average |
|---|---|---|---|---|---|---|---|---|
| Turker Set 1 | 25.224 | 39.860 | 32.444 | 40.625 | 51.049 | 50.000 | 40.964 | 33.227 |
| Turker Set 2 | 30.452 | 46.154 | 35.556 | 42.708 | 53.846 | 51.064 | 46.988 | 37.652 |
| Turker Set 3 | 26.710 | 41.958 | 36.889 | 46.875 | 53.147 | 42.553 | 38.554 | 34.830 |

utilizing GPT-4V as a judger. The process consists of presenting GPT-4V with the question, a corresponding answer generated by the baseline model, and the set of possible options. GPT-4V then assesses whether the model-generated answer is accurate within the given context; Another is open-ended generation where we employ a two-step methodology. We first prompt each model to do open-ended generation. Subsequently, we prompt the model to align its generative response with one of the predefined options: 'a', 'b', 'c', or 'd'.

## B.4 Detailed Results on Multi-faceted Reasoning

In Table 4, we give detailed performance numbers of different MLLMs on multi-faceted reasoning corresponding to Figure 4 in the main paper.

Table 3: Performance of different MLLMs across different disciplines.

| Model | Art&
Sports | Business | Science | Health&
Medicine | Embodied
Tasks | Tech&
Engineering | Average |
|---|---|---|---|---|---|---|---|
| Video-Chat (Open-ended) [4] | 27.484 | 9.091 | 18.137 | 10.417 | 29.371 | 19.149 | 22.887 |
| Video-Chat [4] | 39.355 | 48.951 | 31.863 | 45.833 | 39.161 | 38.298 | 39.588 |
| Video-LLaMA (Open-ended) [12] | 5.419 | 27.972 | 24.020 | 31.250 | 11.816 | 15.957 | 16.096 |
| Video-LLaMA [12] | 27.355 | 31.469 | 31.373 | 48.958 | 16.084 | 28.723 | 28.729 |
| ChatUnivi (Open-ended) [2] | 21.161 | 61.538 | 42.157 | 61.458 | 30.070 | 37.234 | 32.646 |
| ChatUnivi [2] | 12.387 | 58.042 | 50.000 | 60.417 | 30.070 | 43.617 | 29.072 |
| Otter (Open-ended) [3] | 37.677 | 32.867 | 37.255 | 32.292 | 22.378 | 27.660 | 34.639 |
| Otter [3] | 17.677 | 16.783 | 12.255 | 5.208 | 17.483 | 15.957 | 15.876 |
| ImageBind-LLM (Open-ended) [1] | 3.355 | 3.497 | 14.706 | 10.417 | 21.678 | 18.085 | 8.179 |
| ImageBind-LLM [1] | 23.742 | 34.965 | 51.471 | 33.333 | 48.951 | 56.383 | 33.952 |
| PandaGPT (Open-ended) [9] | 22.581 | 16.084 | 24.020 | 21.875 | 19.580 | 21.277 | 21.718 |
| PandaGPT [9] | 27.613 | 44.056 | 39.706 | 25.000 | 40.559 | 21.277 | 31.615 |
| LWM (Open-ended) [6] | 16.000 | 20.979 | 14.706 | 16.667 | 19.580 | 20.213 | 16.976 |
| LWM [6] | 16.387 | 18.182 | 18.137 | 19.792 | 22.378 | 21.277 | 17.938 |
| X-Instruct-BLIP (Open-ended) [8] | 3.613 | 11.888 | 14.706 | 25.000 | 17.483 | 13.830 | 9.416 |
| X-Instruct-BLIP [8] | 19.355 | 13.287 | 22.549 | 29.167 | 18.881 | 14.894 | 19.519 |

Table 4: Detailed results of different MLLMs on multi-faceted reasoning.

| Model | Explanation | Counterfactual
Thinking | Future
Prediction | Domain
Expertise | Attribution
Understanding | Temporal
Understanding |
|---|---|---|---|---|---|---|
| *Proprietary Models* | | | | | | |
| GPT-4V | 44.90 | 64.90 | **78.59** | **61.07** | **59.61** | 27.17 |
| Gemini Pro | **48.58** | **65.49** | 65.45 | 53.87 | 43.92 | 24.65 |
| *Open-source Models* | | | | | | |
| Video-LLaVA-7B | 42.46 | 42.55 | 64.96 | 47.86 | 36.86 | **34.45** |
| VideoChat-7B | 41.66 | 43.73 | 45.74 | 40.95 | 30.59 | 25.77 |
| ImageBind-LLM-7B | 29.51 | 26.86 | 50.61 | 33.93 | 34.90 | 19.89 |
| PandaGPT-7B | 29.55 | 37.45 | 46.47 | 33.93 | 26.27 | 28.01 |
| ChatUnivi-7B | 33.91 | 48.82 | 61.80 | 45.95 | 33.33 | 22.97 |
| VideoLLaMA-2-13B | 10.55 | 23.92 | 25.30 | 16.31 | 8.63 | 6.16 |
| X-Instruct-BLIP-7B | 23.05 | 15.29 | 27.25 | 21.07 | 24.31 | 11.20 |
| LWM-1M-JAX | 11.62 | 18.82 | 30.66 | 17.98 | 21.57 | 7.00 |
| Otter-7B | 16.91 | 10.98 | 15.82 | 13.10 | 17.65 | 9.52 |
| mPLUG-Owl-7B | 35.20 | 49.61 | 55.47 | 47.74 | 24.71 | 20.17 |

## C  Implementation Details

We use the optimum number of video frames and report the performance in the main paper. The numbers of the sampled frames are 10 for GPT-4V/o and Gemini Pro, 8 for Video-LLaVA, 32 for ChatUniVi. For closed-source models, for both Gemini Pro and GPT-4V, we use the default settings provided by their official APIs. We use Katna [2] to extract key video frames as input to these two models. The Gemini Pro is set to process visual input and configured with safety settings to filter a range of harmful content. The configuration thresholds are set to 'BLOCK_NONE'. For PandaGPT, we set 'top_p' to 0.7, and 'temperature' to 0.5. For VideoChat, we set 'max_frames' to 100. For LWM, we use the LWM-Chat-1M variant. For X-Instruct-BLIP, the model is implemented using four image frames. For Otter, we use the video variant. We use GPT-4-32K as the judge for judging whether the model answer is correct when it can not mapped to the option letter using the rule-based method. The prompt provided to GPT-4-32K is structured as follows: "I will present a response from a question-answering model alongside several answer options. Your task is to evaluate the response and determine which of the following options it most closely aligns with, denoting the most similar option by its corresponding letter (a, b, c, or d).".

---

[2]https://github.com/keplerlab/katna

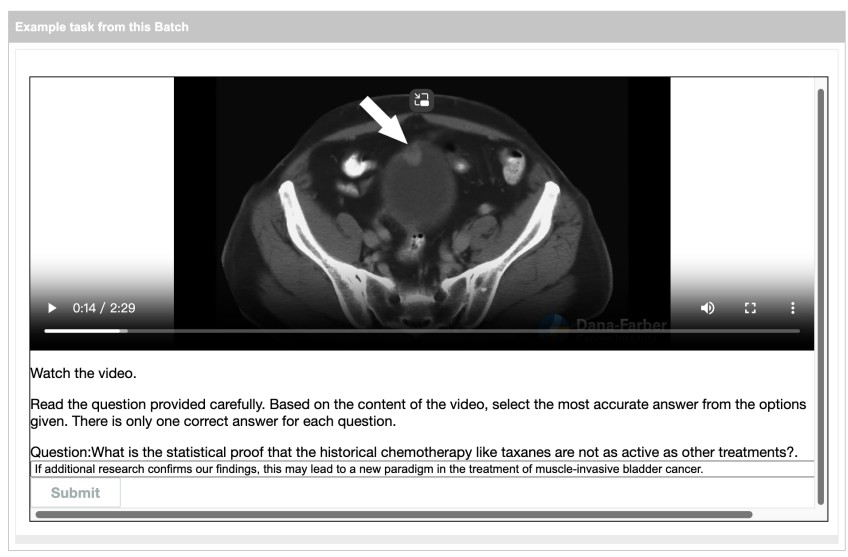

Figure 1: The interface of using Amazon Mechanical Turk to do human evaluation.

Table 5: Category-wise and overall error rates

| Category | Incorrect/Total | Error Rate (%) |
|---|---|---|
| Sports & Arts | 5/62 | 8.06 |
| Health & Medicine | 2/7 | 28.57 |
| Science | 1/52 | 1.92 |
| Robotics | 0/12 | 0.00 |
| Business | 0/10 | 0.00 |
| Tech & Engineering | 1/46 | 2.17 |
| **Overall** | **9/189** | **4.76** |

**Query Generation in Synthetic Data Generation Pipeline** For the discipline of **Science**, queries are generated for subdisciplines such as Geography, Chemistry, Wildlife Restoration, Mycology, Nature, Physics, Weather, Zoology, Math, Botany, Biology, and Geology. In the **Tech & Engineering** discipline, our queries span across Electronics, Animal Behavior, Mechanical Engineering, Energy & Power, Architecture, Agriculture, Nature, Physics, Robotics, Woodworking, and Gardening. The **Sports & Arts** discipline encompasses a broad range of cultural and physical activities, including Music, Drawing and Painting, Football, Volleyball, Aerobic Gymnastics, Basketball, Instrument, Baking, Dance, Woodworking, Graffiti, Anatomy, and additional Music-related topics. **Embodied Tasks** are represented through queries for Assembly, Ego-motion, and Single Object Manipulation, focusing on the interaction between agents and their physical environment. The **Health & Medicine** discipline is segmented into Pharmacy, Public Health, Clinical Medicine, and Basic Medical Science, reflecting the multifaceted nature of healthcare and medical studies. The **Business** discipline is stratified into fundamental areas such as accounting, finance, management, marketing, and economics, each representing key facets of the commercial and economic world. Lastly, the **Game** discipline consists of Role Playing Game, First Person Shooting game, Racing Game, Adventure Game, Real-Time Strategy Game, Tower Defense game, and Fighting Game.

Each generated query retrieves relevant video content, which is then filtered and processed to align with the specific needs of our research objectives. Videos that meet our criteria in terms of content, length, and quality are downloaded and incorporated into our dataset, forming the basis for subsequent analysis and model training.

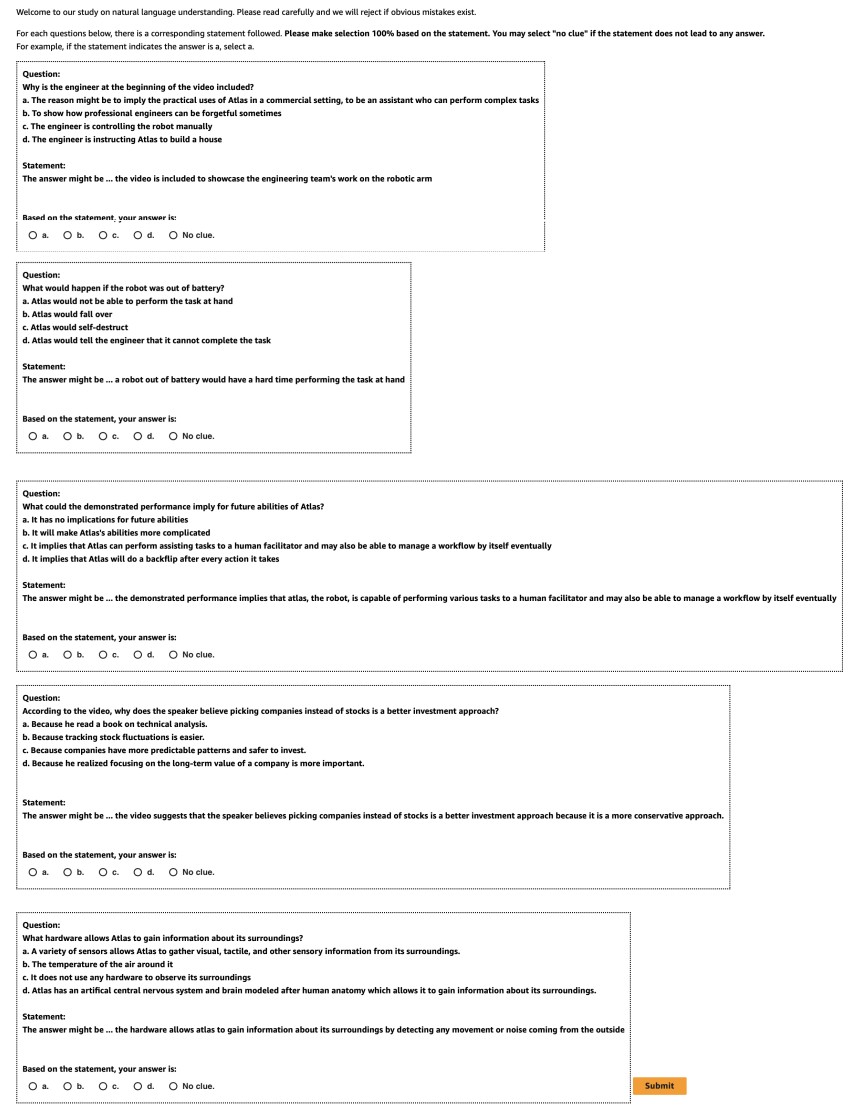

Figure 2: Human evaluation interface for GPT judger.

## D Human Evaluation

### D.1 Quality of Data

We hired Amazon Mechanical Turk to do human evaluation on the data with the results shown in Table 2. Workers were required to have completed more than 1000 Human Intelligence Tasks (HITs) and have an HIT approval rate greater than 95% to qualify for our tasks. We show in Figure 1 the human evaluation interface on the generated data. Each worker was compensated 0.20 for completing an assignment. This amount was determined based on the estimated time and effort required to complete each task. We set the number of unique workers per task to 3 to collect diverse perspectives while avoiding redundancy. Workers were given 1 hour to complete each assignment. This time frame was chosen to enable thoughtful responses from workers.

We also hired students from campus to do human evaluation on subset of the data. The results are shown in Table 6. The performance of the human evaluators did not surpass that of GPT-4V and Gemini-Pro. This outcome underscores the challenging nature of the dataset, which often necessitates specialized domain knowledge that our evaluators—primarily non-experts—found demanding. These

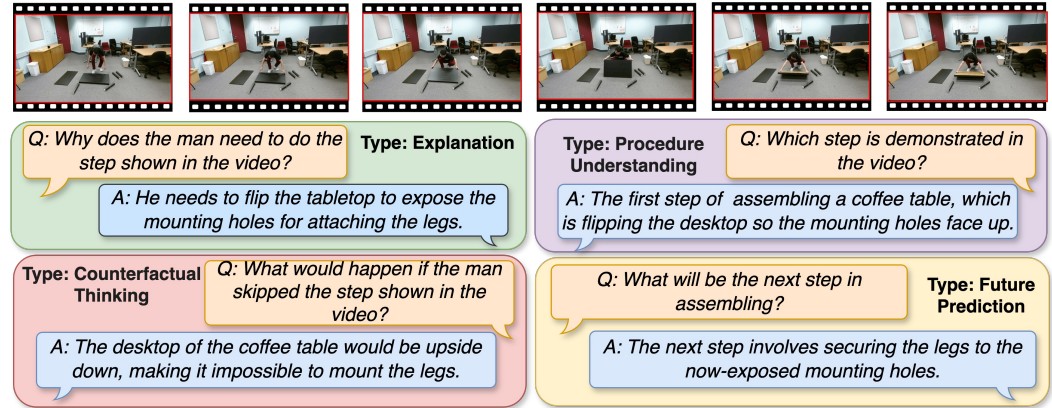

Figure 3: Examples from MMWorld in the Embodied Tasks discipline.

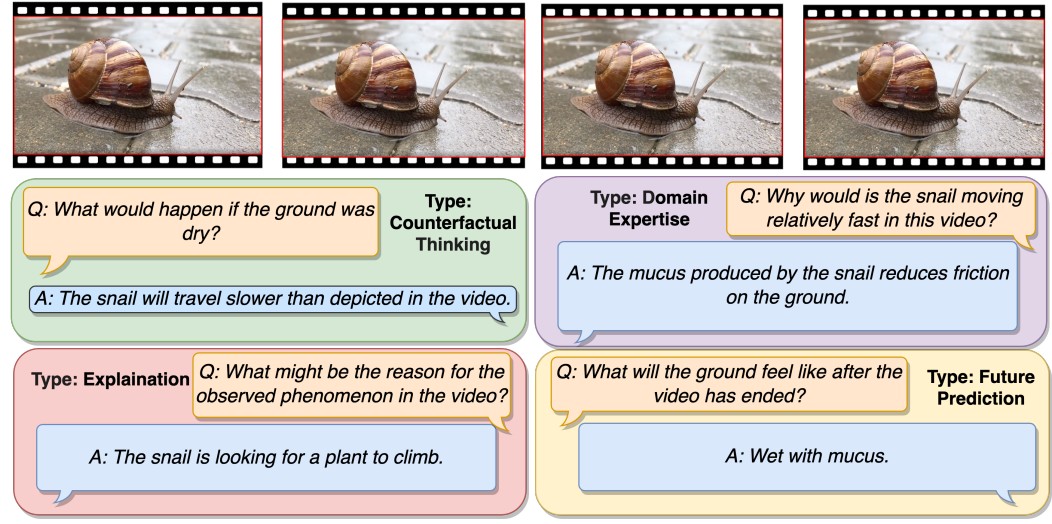

Figure 4: Examples from MMWorld in the Tech & Engineering discipline.

results highlight the complexity of the questions and the potential necessity for discipline-specific understanding to achieve high accuracy

## D.2  Quality of Using GPT as the Judger

For a comprehensive assessment of GPT-4V's accuracy when using it as the judger, we devised a human evaluation protocol also resort to Amazon Mechanical Turk, as visualized in Figure 2. The evaluators present a series of statements derived from the video, and GPT-4V is tasked with selecting the most accurate answer from a set of multiple-choice questions. Through this interface, human evaluators can efficiently gauge GPT-4V's performance across different types of questions—when using it as the judger.

The results obtained from this human evaluation process are shown in Table 5, across 189 examples, there are only 9 incorrect ones with the error rate of 4.76%, validating the effectiveness of using GPT-4V as the judger.

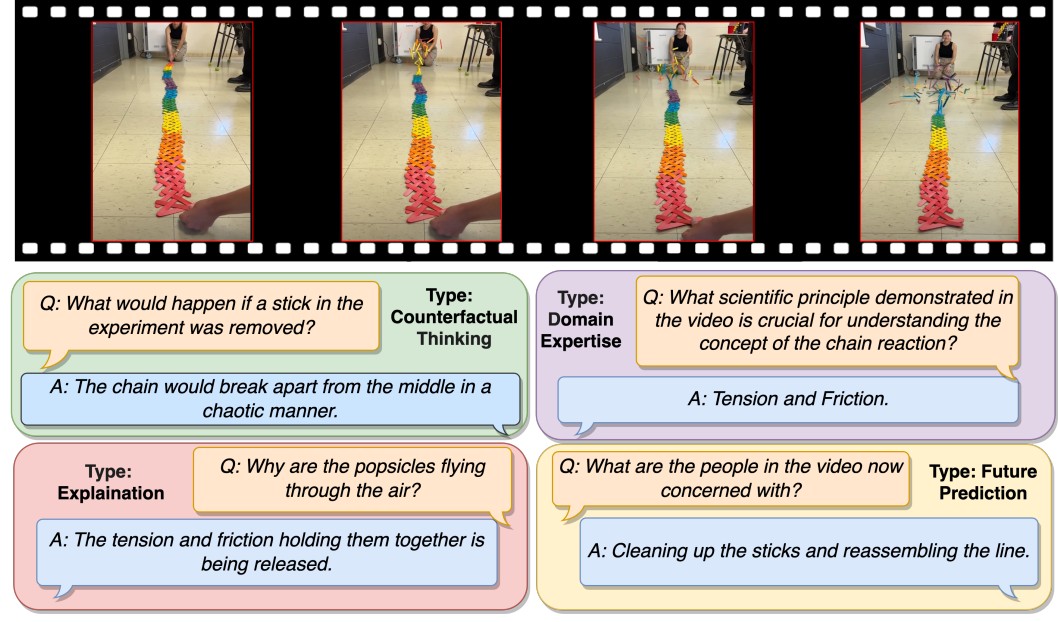

Figure 5: Examples from MMWorld in the Science discipline.

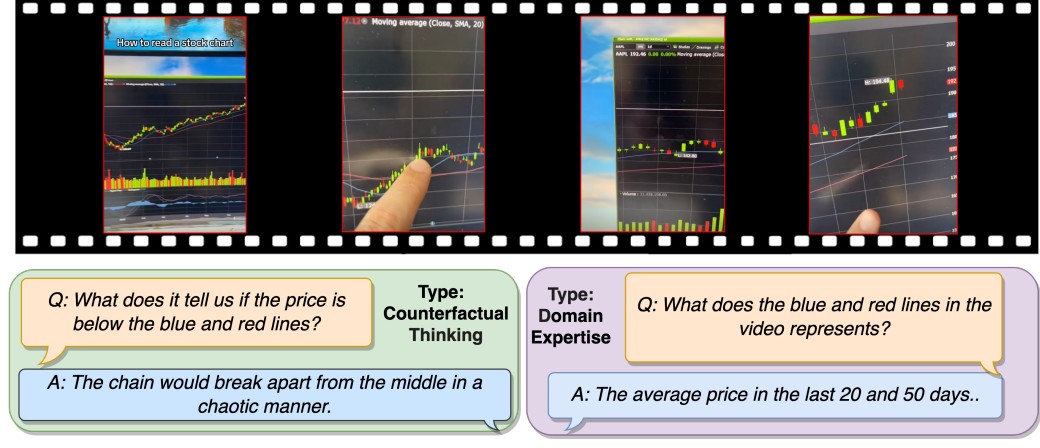

Figure 6: Examples from MMWorld in the Business discipline.

# E Error Analysis

In this section, we delve into the analysis of errors from evaluated MLLMs. We summarized error types as follows:

*Question Understanding Error (QUE):* Models misinterpret the question's intent, such as misunderstanding how a pendulum's period would change if a condition in the scenario is altered.

*Audio Understanding Error (AUE):* Models fail to interpret audio cues correctly, shown by their failure to recognize blue and red lines on a stock chart.

*Visual Perception Error (VPE):* There is a misinterpretation of visual content, leading to incorrect assumptions about the visual data presented in the video.

*Hallucinations (HE):* Models generate content or details that are not present in the actual data, essentially 'hallucinating' information.

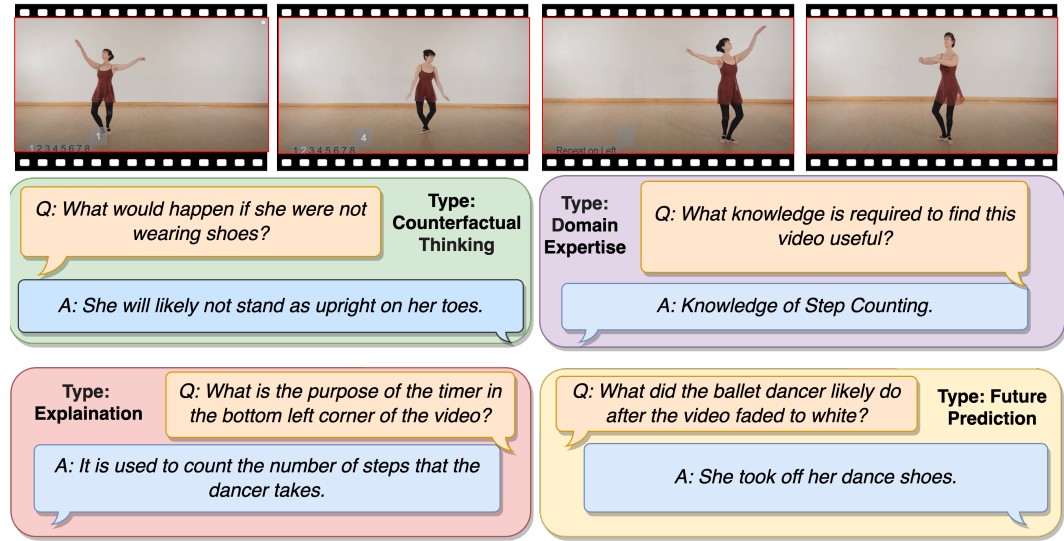

Figure 7: Examples from MMWorld in the Arts & Sports discipline.

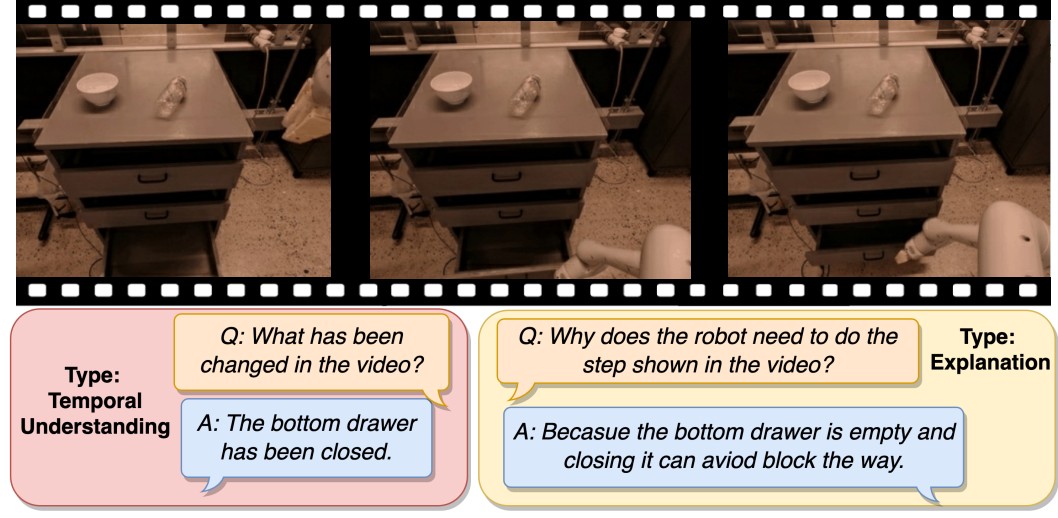

Figure 8: Examples from MMWorld of explicit temporal understanding and implicit temporal understanding (e.g., in explanation).

*Reasoning Error (RE):* Models demonstrate a lack of logical reasoning, leading to incorrect conclusions based on the given data.

*Lack of Domain Knowledge (LDK):* Models show an inability to answer questions that require specific domain expertise, indicating a gap in their knowledge.

*Reject to Answer (RA):* An example of this error was observed when the model was asked to select an answer regarding the outcome of an experiment involving liquid nitrogen. Instead of choosing an option, the model provided an unrelated response concerning a light bulb, indicating either a misunderstanding or a cautious approach due to the potential for the question to be interpreted as pertaining to a sensitive topic, which can trigger content filters focused on safety and compliance policies.

Table 6: Comparison of Human Evaluation on subset of 75 videos.

| Model | Art&Sports | Business | Science | Health&Medicine | Embodied Tasks | Tech&Engineering | Average |
|---|---|---|---|---|---|---|---|
| **Human Evaluation** | **31.183** | **59.782** | **42.103** | **48.858** | **56.429** | **50.134** | **43.758** |
| GPT-4V [7] | 30.399 | 89.203 | 68.731 | 80.059 | 38.432 | 69.108 | 48.793 |
| Gemini-Pro [10] | 28.745 | 80.909 | 69.425 | 80.023 | 50.987 | 80.479 | 48.083 |

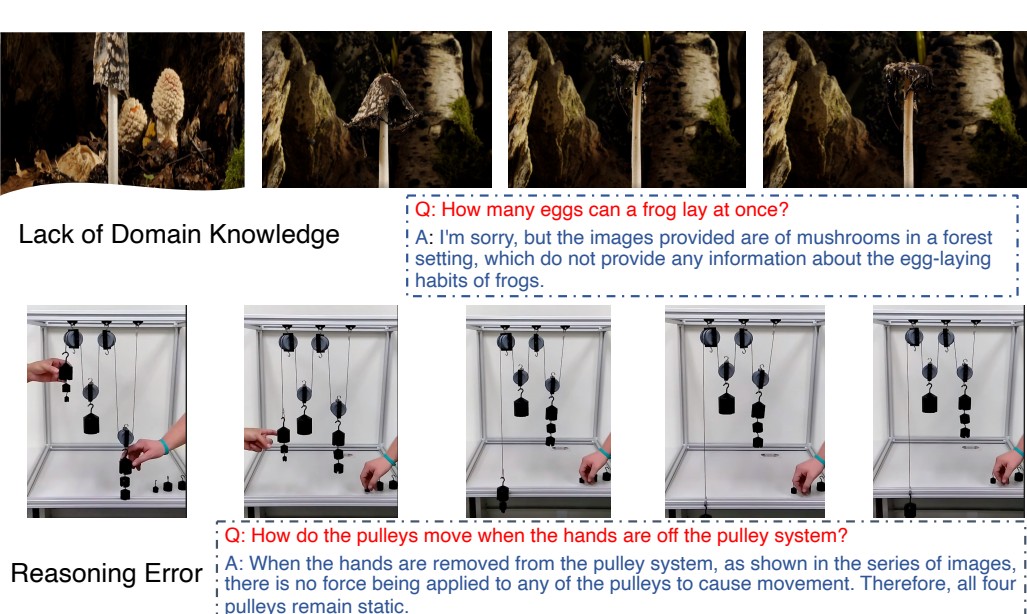

Figure 9: Error pattern of lack of domain knowledge and reasoning error. In the first case, the model does not give the correct answer because of lacking the domain knowledge. In the second case, the model makes the wrong reasoning.

We show in Figure 9, 10, 11, 12 some error cases of *Question Understanding Error*, *Audio Understanding Error*, *Visual Perception Error*, *Hallucinations*, *Reasoning Error*, *Lack of Domain Knowledge*, and *Reject to Answer* respectively from MLLMs evaluated on MMWorld.

## F  Data Examples

We show in Figure 3, 4, 5, 6, 7, 8 some additional examples from MMWorld.

## G  Additional Data Statistics

For human annotated dataset, the length of each video was capped at approximately two minutes. The statistical distribution of the disciplines within the dataset for this part is as follows:

- *Sports & Arts*: The subset that consists of 77 videos, showcasing a vibrant collection that covers a wide range of topics from athletic endeavors to various forms of artistic expression.

- *Science*: A subset of 75 videos, which delves into the empirical world of scientific inquiry, spanning a multitude of specializations from fundamental physics to advanced biological studies.

- *Tech & Engineering*: Encompassing 54 videos, this segment captures the cutting-edge advancements and foundational concepts that drive innovation and infrastructure in the modern world.

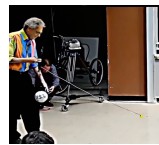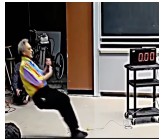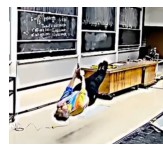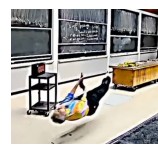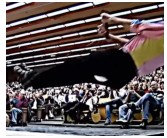

Question Understanding Error

> Q: What would happen to the pendulum period if the professor sit up straight on the ball instead?
>
> A: The pendulum period would not afford it.

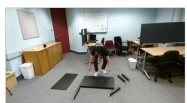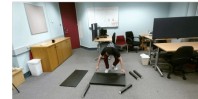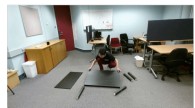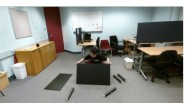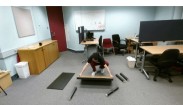

Visual Perception Error, Hallucinations

> Q: What would happen if the man skipped the step shown in the video?
> A: If the man skipped the step of rolling the blue tube on the floor to uncover the red box, he would likely have encountered difficulty in accessing the contents of the box. This step is important because it helps the man to create a flat surface to work with the red box, ensuring that the box is visible and accessible.

Figure 10: Error pattern of question understanding, visual perceptron error, and hallucinations. In the first case, the model does not understand the question correctly where the model fails to accurately discern the query regarding the pendulum's period. In the second scenario, the model erroneously identifies objects within the visual input, leading to the hallucination of non-existent elements, such as a red box.

- *Embodied Tasks*: With 50 videos, the dataset provides a focused insight into the dynamic field of Embodied Tasks, highlighting the intersection of AI, mechanics, and automation.

- *Health & Medicine*: This essential discipline is well-represented with 50 videos, offering perspectives on medical breakthroughs, healthcare practices, and life sciences.

- *Business*: This discipline includes 50 videos, reflecting on the multifaceted nature of commerce, from economics to management sciences.

- *Game*: This discipline includes 51 videos, reflecting various aspects of gaming.

Altogether, the MMWorld Benchmark's diversity is visually encapsulated in Figure 13, which delineates the distribution of videos across 61 subdisciplines. The horizontal bar chart provides a quantified representation of the dataset's range, reflecting the careful curation process that has gone into ensuring breadth across various knowledge areas.

The world we live in is rich with both audio and visual information, and effective world modeling requires an understanding of how these modalities interact and convey meaning. To achieve this, we annotated additional attributes such as "Requires Audio," "Requires Video," and "Question Only." These annotations help determine whether correctly answering a question necessitates audio information, visual cues from the video, or can be addressed based solely on the question itself. By doing so, we ensure that our benchmark tests the full spectrum of multimodal comprehension, reflecting the complex, sensory-rich environment in which real-world understanding takes place. The statistics of these annotations are shown in Figure 14.

# H Datasheets

## H.1 Motivation

**For what purpose was the dataset created?**

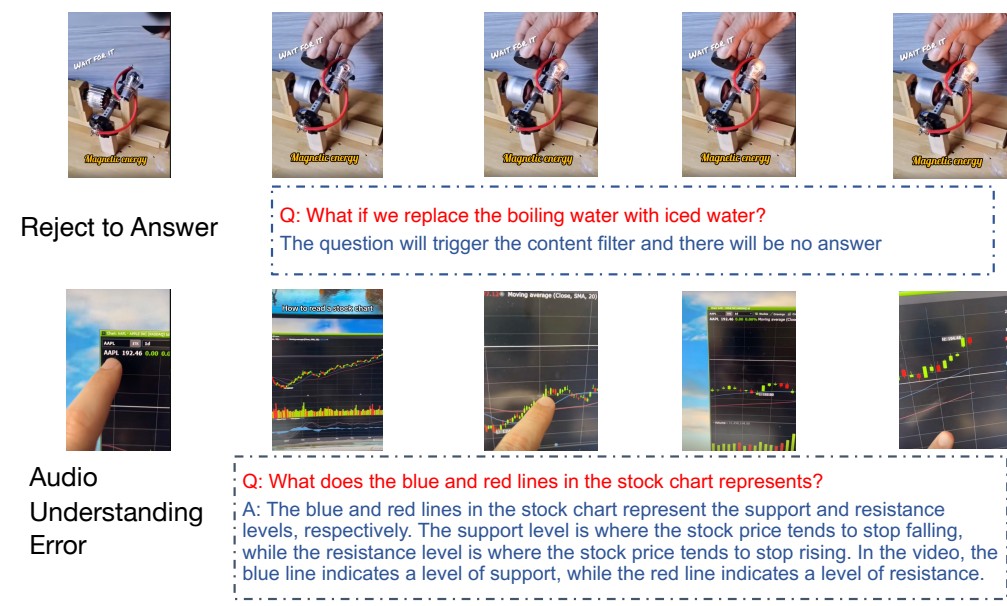

Figure 11: Error pattern of reject to answer and audio understanding error. In the first case, the model will reject to answer because of the dangerous action showing in the video. In the second case, the model does not understand the audio correctly as the answer to the question is given in the audio of the video.

To introduce a multi-discipline multi-faceted multimodal video understanding benchmark to comprehensively evaluate MLLMs' abilities in reasoning and interpreting real-world dynamics.

**Who created the dataset (e.g., which team, research group) and on behalf of which entity (e.g., company, institution, organization)?**

The dataset is created by authors from UCSC, UCSB, and Microsoft.

**Who funded the creation of the dataset?**

UCSC, UCSB, and Microsoft Azure.

## H.2   Composition

**What do the instances that comprise the dataset represent? (e.g., documents, photos, people, countries)**

Videos along with captions and question/answer pairs.

**How many instances are there in total (of each type, if appropriate)?**

6,627 instances. The data distribution over different types can be found in Figure 2 of the main paper.

**Does the dataset contain all possible instances or is it a sample (not necessarily random) of instances from a larger set?**

Yes.

**Is there a label or target associated with each instance?**

Yes.

**Is any information missing from individual instances?**

No.

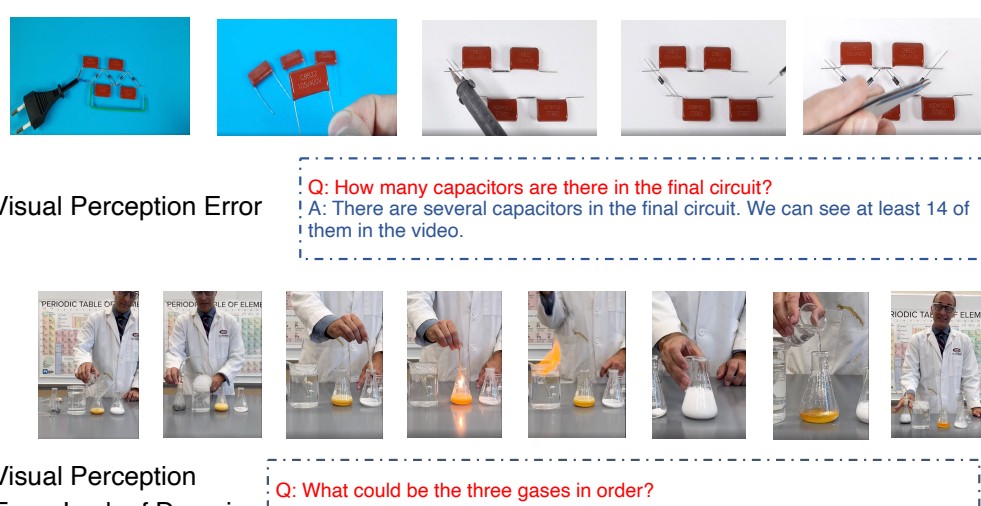

Figure 12: Error pattern due to visual perception inaccuracies and insufficient domain knowledge. The first case demonstrates a visual perception error where the model incorrectly identifies the number of capacitors present. The second case showcases a compound error where the model not only fails to discern the colors indicative of different gases but also lacks the domain knowledge necessary to infer their identity correctly.

**Are relationships between individual instances made explicit (e.g., users' movie ratings, social network links)?**

N/A.

**Are there recommended data splits (e.g., training, development/validation, testing)?**

The MMWorld is used for evaluation purpose only.

**Are there any errors, sources of noise, or redundancies in the dataset?**

No.

**Is the dataset self-contained, or does it link to or otherwise rely on external resources (e.g., websites, tweets, other datasets)?**

Yes.

**Does the dataset contain data that might be considered confidential?**

No.

**Does the dataset contain data that, if viewed directly, might be offensive, insulting, threatening, or might otherwise cause anxiety?**

No.

### H.3 Collection Process

The data collection process is described in Section 3 of the main paper.

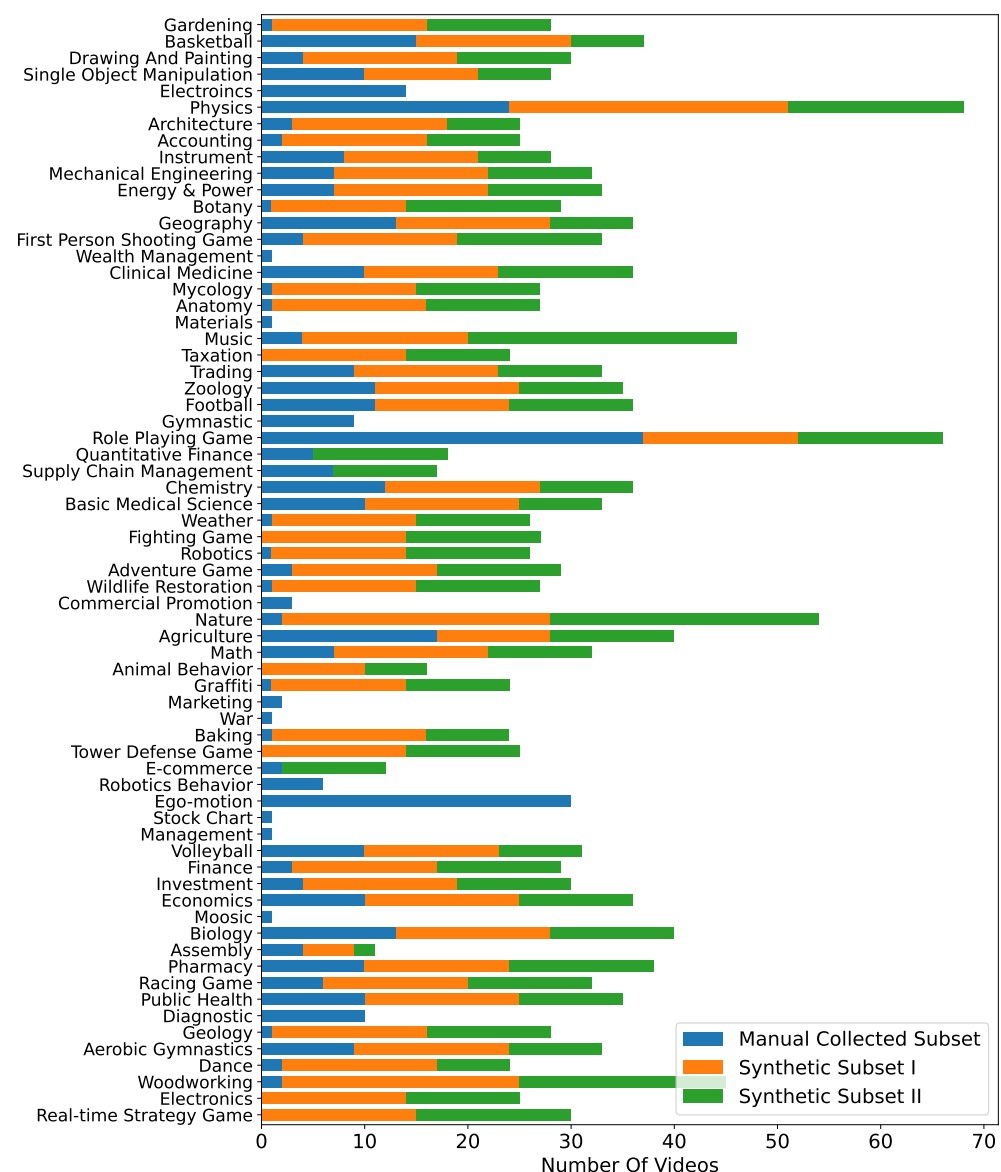

Figure 13: The number of videos per subdiscipline in MMWorld. Each horizontal bar indicates the quantity of videos corresponding to a subdiscipline, showcasing the dataset's diversity and coverage across various domains of knowledge. Synthetic Subset I is collected with audio-only data and Synthetic Subset II is collected with visual-only data.

### H.4 Preprocessing/cleaning/labeling

**Was any preprocessing/cleaning/labeling of the data done (e.g., discretization or bucketing, tokenization, part-of-speech tagging, SIFT feature extraction, removal of instances, processing of missing values**

We extract video frames from collected videos in automatically generated.

**Was the "raw" data saved in addition to the preprocessed/cleaned/labeled data (e.g., to support unanticipated future uses)?**

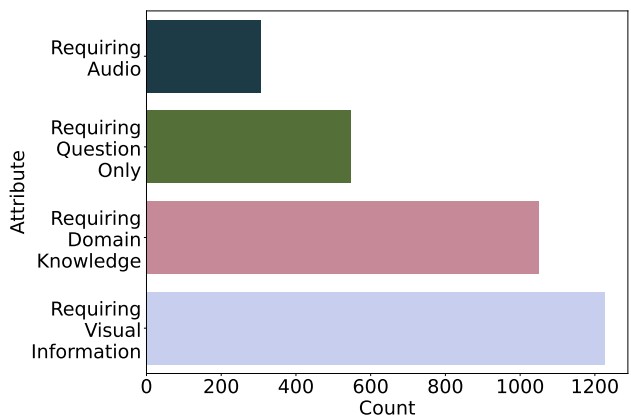

Figure 14: The distribution statistics of questions in the MMWorld benchmark by annotations.

Yes. The raw video urls are given.

**Is the software that was used to preprocess/clean/label the data available?**

Yes. The source code can be found in `https://github.com/eric-ai-lab/MMWorld`.

### H.5 Uses

**Has the dataset been used for any tasks already?**

Yes. We have used the dataset to evaluate video question answering.

**Is there a repository that links to any or all papers or systems that use the dataset?**

Yes. The GitHub repository `https://github.com/eric-ai-lab/MMWorld` here.

**What (other) tasks could the dataset be used for?**

Video captioning and evaluating faithfulness of evaluation metrics.

**Is there anything about the composition of the dataset or the way it was collected and preprocessed/cleaned/labeled that might impact future uses?**

No.

**Are there tasks for which the dataset should not be used?**

The videos in this dataset are from different sources and are unique. The dataset should not be used for tasks such as video editing.

### H.6 Distribution

**Will the dataset be distributed to third parties outside of the entity (e.g., company, institution, organization) on behalf of which the dataset was created?**

Yes. The benchmark is publicly available.

**How will the dataset will be distributed (e.g., tarball on website, API, GitHub)?**

We host it on the webpage, GitHub, and Huggingface.

**When will the dataset be distributed?**

It's availale and open to the public now.

**Will the dataset be distributed under a copyright or other intellectual property (IP) license, and/or under applicable terms of use (ToU)?**

CC-By 4.0.

**Have any third parties imposed IP-based or other restrictions on the data associated with the instances?**

No.

**Do any export controls or other regulatory restrictions apply to the dataset or to individual instances?**

No.

### H.7 Maintenance

**Who will be supporting/hosting/maintaining the dataset?**

The authors will be supporting/hosting/maintaining the dataset.

**How can the owner/curator/manager of the dataset be contacted (e.g., email address)?**

The email address is xhe89@ucsc.edu.

**Is there an erratum?**

No. We will make it if there is any erratum.

**Will the dataset be updated (e.g., to correct labeling errors, add new instances, delete instances)?**

Yes. We will make announcements on GitHub if there is any update.

**If the dataset relates to people, are there applicable limits on the retention of the data associated with the instances (e.g., were individuals in question told that their data would be retained for a fixed period of time and then deleted)?**

N/A.

**Will older versions of the dataset continue to be supported/hosted/maintained?**

Yes. Old versions can still be accessed from Huggingface.

**If others want to extend/augment/build on/contribute to the dataset, is there a mechanism for them to do so?**

Yes. Contributors can post issues or submit pull requests on GitHub. We will review and verify contributions, and update the dataset if the contribution is useful.

## I   Author Statement, Hosting, Licensing, and Maintenance Plan

**Author Statement**   We bear all responsibility in case of violation of rights and confirmation of the data license.

**Hosting**   MMWorld is hosted on `https://mmworld-bench.github.io/`. The dataset is provided in the JSON file format. The metadata can be found at `https://huggingface.co/datasets/Xuehai/MMWorld`.

**License**   MMWorld is licensed under the CC-BY 4.0 license.

**Maintenance Plan**   We will keep maintaining and updating the dataset and benchmark, including the leaderboard.