# OpenReview forum: "MMWorld: Towards Multi-discipline Multi-faceted World Model Evaluation in Videos"
_NeurIPS.cc/2024/Datasets_and_Benchmarks_Track — Submitted to NeurIPS 2024 Track Datasets and Benchmarks_

### Official Review · Reviewer_7qvY · 2024-07-22

**Rating:** 4
**Confidence:** 5
**Correctness:** Yes
**Clarity:** Yes

**Review:**

Strengths:

1. MMWorld spans a wide range of disciplines and tasks, offering a thorough evaluation of MLLMs’ capabilities in understanding and reasoning about real-world dynamics across different domains.
2. The benchmark includes diverse question types that require multi-faceted reasoning, providing insights into models’ strengths and weaknesses in areas like explanation, counterfactual thinking, and future prediction.
3. The study offers an in-depth analysis of MLLMs’ performance, including comparisons with human performance, which helps identify specific areas where models excel or struggle, guiding future improvements

Weaknesses:

1. This study views MLLMs as world models, claiming that these models interpret and reason about complex world dynamics. However, (multimodal) LLMs take in multimodal inputs and generate language-only outputs. The term "world model" may not be the most accurate description of the model's capabilities, especially since it is often used in the context of reinforcement learning and planning, which involves modeling future states of the environment, such as images, videos, or other latent representations/transitions. These world models provide agents with scalable supervision signals, but multimodal LLMs are not adaptable for these tasks. Therefore, can the authors provide a more accurate term to describe the multimodal LLMs' capabilities?

2. In experiments, such as in Table 3, the evaluation is based on multiple-choice questions, which may not fully capture the model's understanding of world dynamics.
     - How does the paper measure the model's understanding of world dynamics? Is the MLLM a better "world model" choice (even if we determine MLLM is a world model) than text-vision models (e.g., Dreamer V3, Copilot4D, PolyGRAD)?
     - The differences between the multiple-choice questions in this paper and multimodal QA reasoning tasks in prior papers should be discussed, particularly what makes this paper a true "world model evaluation."

**Strengths:**

See _Review_

**Additional Feedback:**

N.A.

**Documentation:**

Yes

**Ethics:**

N.A.

**Limitations:**

N.A.

**Opportunities For Improvement:**

See _Review_

**Relation To Prior Work:**

Yes

**Summary And Contributions:**

The paper introduces MMWorld, a new benchmark designed to evaluate the world modeling capabilities of Multimodal Language Models (MLLMs) using videos that span seven broad disciplines and 69 subdisciplines. MMWorld focuses on multi-faceted reasoning tasks such as explanation, counterfactual thinking, and prediction. The benchmark highlights areas where MLLMs need improvement and shows how these models’ performance and skill sets differ from human capabilities.

---

> ### Author Rebuttal · Authors · 2024-08-17
>
> Thank you for your thoughtful review and for providing detailed feedback. We appreciate your insights and have carefully considered your comments. Below, we provide our pointwise responses to your comments:
>
> >- This study views MLLMs as world models, claiming that these models interpret and reason about complex world dynamics. However, (multimodal) LLMs take in multimodal inputs and generate language-only outputs. The term "world model" may not be the most accurate description of the model's capabilities, especially since it is often used in the context of reinforcement learning and planning, which involves modeling future states of the environment, such as images, videos, or other latent representations/transitions. These world models provide agents with scalable supervision signals, but multimodal LLMs are not adaptable for these tasks. Therefore, can the authors provide a more accurate term to describe the multimodal LLMs' capabilities?
>
> We would like to clarify that in our paper, we extend "world model" from its traditional use in reinforcement learning (RL) to a **generalized sense**. To effectively model the world, one must first understand it. Our broader definition of a world model includes multi-discipline and multi-faceted reasoning abilities. While the traditional RL definition involves modeling future states of the environment, MLLMs also has the potential to serve as world models through language-based representations. For example, MLLMs can generate code in symbolic representations, which allows for formal planning or inference in a neurosymbolic context. MLLMs **with world model capabilities** should be able to predict future states and contribute to decision-making through language-based representations;
>
> As LeCun et al. [28] discussed, the world model should be able to (1) estimate missing information about the state of the world not provided by perception, and (2) predict plausible future states of the world. Evaluation of such capabilities requires multi-faceted reasoning beyond perception level, and therefore we build MMWorld with multi-faceted reasoning questions. In MMWorld, 8.8% questions are in “Future Prediction” type, where models are required to predict future scenarios in the language form.
>
> We also want to emphasis that our benchmark is **not a sufficient but an essential and necessary testbed** for world model evaluation (as stated in the Intro). While LLMs exhibit emerging capabilities related to "world models," our intention is **not to suggest** that they can already serve as world models. Rather, our benchmark is designed to serve as an indicator, **evaluating their potential** in the context of world modeling. We believe overcoming the unique challenges presented in MMWorld is essential and necessary towards comprehensive world modeling.  We will continue to update the MMWorld benchmark, adding new data and incorporating new models.
>
> >- In experiments, such as in Table 3, the evaluation is based on multiple-choice questions, which may not fully capture the model's understanding of world dynamics.
>
> We adopted multiple-choice questions in MMWorld because they are easier to quantify compared to open-ended generation tasks. Open-ended generation evaluation presents challenges due to its subjectivity, whereas multiple-choice questions offer more straightforward, objective assessment; Multiple-choice questions are also a relatively reliable evaluation method for multimodal video understanding tasks, which often require complex reasoning across time and space. These questions can be carefully designed to target specific aspects of this reasoning, such as temporal understanding or different facets of reasoning, allowing for focused and controlled evaluation with a limited set of candidate answers; Additionally, the quality of our multiple-choice questions has been validated by three sets of Amazon Turk workers and highly educated human experts. To further ensure the reliability of our evaluations, we conducted a Random Choice baseline (shown in Table 3) and applied circular evaluation during human evaluation to mitigate option position bias.
>
> Meanwhile, we recognize the value of open-ended generation, and we have now conducted additional experiments with open-ended generation. In these experiments, we presented models with questions and videos and asked them to generate answers directly. We then used GPT-4o to rank the answers from the other 11 evaluated models in our main paper, comparing them to the ground truth, with GPT-4o outputting the ranking of each model for each question.
>
>
> The average rankings across all questions are as follows:
> | Model               | Ave Ranking |
> |---------------------|---------|
> | Gemini Pro          | 1.08    |
> | Video-LLaVA-7B      | 2.21    |
> | Video-Chat-7B       | 3.34    |
> | mPLUG-Owl-7B        | 3.69    |
> | ChatUnivi-7B        | 4.01    |
> | PandaGPT-7B         | 6.12    |
> | ImageBind-LLM-7B    | 6.41    |
> | X-Instruct-BLIP-7B  | 8.27    |
> | LWM-1M-JAX          | 9.03    |
> | Otter-7B            | 10.55   |
> | Video-LLaMA-2-13B   | 10.88   |
>
>
> The results showed a strong correlation with the rankings from the multiple-choice questions in Table 3, with only ChatUnivi-7B and mPLUG-Owl-7B swapping positions. This correlation reinforces the reliability of our multiple-choice evaluation.
>
>
> [28] A path towards autonomous machine intelligence version

---

> > ### Comment · Area_Chair_2SMW · 2024-08-29
> > **Reminder to response to author rebuttal**
> >
> > Dear Reviewer,
> >
> > The ddl for author and reviewer discussion is approaching. Please check the author rebuttal and leave some comments to respond to author rebuttal.
> >
> > Thanks,
> >
> > Your AC

---

> ### Author Rebuttal · Authors · 2024-08-17
>
> >- How does the paper measure the model's understanding of world dynamics? Is the MLLM a better "world model" choice (even if we determine MLLM is a world model) than text-vision models (e.g., Dreamer V3, Copilot4D, PolyGRAD)?
>
> MMWorld measures the model’s understanding from multi-faceted reasoning tasks, including future prediction, counterfactual reasoning, etc, across various disciplines. As described in Sec 3.1, "Manual Data Collection" of our paper, MMWorld benchmark also emphasizes temporal information (the principle when we collecting the dataset), prioritizing videos that provide meaningful content over time. The evaluated models need to do temporal reasoning, which is essential for grasping world dynamics.
>
> Our paper focuses on **benchmarking and evaluation** rather than proposing a better “world model.” The motivation behind MMWorld is to evaluate the multi-disciplinary, multi-faceted reasoning abilities of MLLMs and assess their potential as world models. Both MLLMs and text-vision models have their strengths and limitations. Text-vision models like Dreamer V3, Copilot4D, and PolyGRAD excel in predicting the outcome of a sequence of actions in very specific domain, while MLLMs have commonsense and domain knowledge and broad generalizability; In addition, MLLMs were pretrained by large scale of diverse data, they have multi-discipline world knowledge and are good at multi-facet reasoning such as inverse dynamic prediction—predicting intermediate steps between given previous and next states. This capability is useful as an auxiliary task for next-state prediction [1][2][3] and important in real-world reasoning.
>
> Specifically, Dreamer V3 proposes an algorithm training model sizes ranging from 12M to 400M parameters, making RL readily applicable to new problems. Their domains are for robot locomotion and manipulation task and games. While MLLMs usually have larger number of parameters equipped with larger capacity that have domain knowledges in differen domains;  Copilot4D proposes a novel world modeling approach that first tokenizes sensor observations with VQVAE, then predicts the future via discrete diffusion. It is based on diffusion model and lacks the multi-discipline domain knowledge found in MLLMs; PolyGRAD proposes the first approach that enables on-policy trajectory generation without autoregressive sampling and can generate entire on-policy trajectories in a single pass through a diffusion model using the MuJoCo environments. While effective in their respective domains, these models are more specialized compared to the general world knowledge and reasoning capabilities of MLLMs.
>
> Overall, we believe that both RL-based models and MLLMs have their unique strengths and are complementary. The MMWorld benchmark is designed to measure a generalist model’s multi-disciplinary domain knowledge, with a particular focus on multi-faceted reasoning, including inverse dynamic reasoning across diverse domains. We will keep updating the benchmark to add new models and data.
>
>
> [1] BERT: Pre-training of Deep Bidirectional Transformers for Language Understanding
>
> [2] ViLBERT: Pretraining Task-Agnostic Visiolinguistic Representations for Vision-and-Language Tasks
>
> [3] Planning from Pixels using Inverse Dynamics Models
>
> >- The differences between the multiple-choice questions in this paper and multimodal QA reasoning tasks in prior papers should be discussed, particularly what makes this paper a true "world model evaluation."
>
> As discussed and shown in Table 1 of the paper, MMWorld distinguishes itself from previous multimodal QA reasoning benchmarks with two unique advantages: (1) multi-discipline, covering various disciplines that often require domain expertise for comprehensive understanding; (2) multi-faceted reasoning, including explanation, counterfactual thinking, future prediction, etc. Our multiple-choice questions are carefully designed and manually annotated to test these multi-faceted reasoning abilities across various disciplines, making them particularly effective for evaluating the diverse capabilities of MLLMs. These two aspects are important for world model evaluation.
>
> We have made every effort to clarify and address your concerns in our responses. Your insights have been valuable, and we truly appreciate them. If you have any further questions or need additional clarification, please don't hesitate to let us know, and we will try our best to provide further explanations. Thank you again for your constructive feedback!

---

> ### Author Response · Authors · 2024-08-28
>
> Dear Reviewer 7qvY,
>
> Thank you once again for your valuable reviews, comments, and suggestions! As the discussion period nears its end, we want to ensure that we've adequately addressed all of your concerns. If you have any further comments or questions, please don't hesitate to let us know. We will do our best to clarify any remaining points within the remaining time.
>
> Thank you for your continued engagement.
>
> Best regards,
>
> The Authors

---

### Official Review · Reviewer_YWxW · 2024-07-22
**This paper proposes a comprehensive benchmark for measuring the ability of video-LLMs to understand the world, ingeniously leveraging the classification and data from the YouTube-8M dataset, with a complete data collection process and experimental analysis.**

**Rating:** 8
**Confidence:** 4
**Correctness:** Seems correct.
**Clarity:** Clearly written.

**Review:**

The quality of this work is high, with clear figures and comprehensive considerations. Table 1 and Figure 2 clearly demonstrate the construction and components of MMWorld. Figure 3 illustrates the data collection process.

In terms of result analysis, the paper presents most of the data results in Table 3 and provides a lot of valuable information in subsequent analyses and supplementary materials, including samples and error analysis.

**Strengths:**

1.Clear figures and smooth writing.

2.The design and construction of the dataset are professional.

3.The validation and analysis of related methods are thorough.

**Additional Feedback:**

It is best to devise a way to confirm that benchmark's questions need to be answered in combination with video, rather than through general knowledge stored in pretrained LLMs.

**Documentation:**

Seems sufficient.

**Ethics:**

No.

**Limitations:**

See the "Opportunities for Improvement" section above.

**Opportunities For Improvement:**

1.Extracting video key frames to obtain QA using GPT4V may face two issues: (1) it cannot guarantee that these QAs are derived from GPT's stored common sense or based on the video content obtained; (2) the paper mentions Video-ChatGPT data in the explanation of related methods but does not test it, which is strange.

2.The current method still treats videos as key frame collections for solving problems. Is it possible to examine some video-specific information, such as instance consistency issues and more temporal-related problems?

**Relation To Prior Work:**

With a clear table comparison.

**Summary And Contributions:**

1.This paper ingeniously combines the world model with video-LLMs and for the first time introduces the concept of multidisciplinarity.

2.MMWorld distinguishes itself from previous video understanding benchmarks with two unique advantages: (1) multidisciplinarity, covering a variety of disciplines that require domain expertise for comprehensive understanding; (2) multifaceted reasoning, including explanation, counterfactual thinking, and future prediction.

3.The capacity of MMWorld is substantial, comprising 1,910 videos across seven broad disciplines and 69 sub-disciplines, with a total of 6,627 question-answer pairs and their related content. The evaluation includes 2 proprietary and 10 open-source MLLMs.

---

> ### Author Rebuttal · Authors · 2024-08-17
>
> Thank you for your positive review! We’re pleased that you found the quality of our work high and appreciated the clarity of our presentations and thoroughness of our analyses. Below, we provide pointwise responses to your suggestions:
>
> >- Extracting video key frames to obtain QA using GPT4V may face two issues: (1) it cannot guarantee that these QAs are derived from GPT's stored common sense or based on the video content obtained; (2) the paper mentions Video-ChatGPT data in the explanation of related methods but does not test it, which is strange.
>
>
> (1) We appreciate your feedback and would like to clarify that the primary focus of MMWorld's evaluation is on human-annotated data. All videos in the benchmark were carefully watched and annotated by human annotators. GPT-4V was not used to extract video key frames during this process. The main results presented in the paper are derived from this human-annotated data, ensuring that the evaluations are grounded in a reliable and accurate understanding of the video content.
>
> Additionally, we conducted an evaluation on synthetic data using text-only inputs. The results show a significant drop in performance when video content is excluded: the evaluated models achieved 40.27% accuracy on Synthetic Subset II (Visual) when video content was included, but only 13.95% without it. This further emphasizes the importance of video content in generating accurate question-answer pairs, ensuring that the QA is based on the video and not just general knowledge stored in the models.
>
>
>
>
> (2) Regarding the inclusion of Video-ChatGPT data, we have now expanded our evaluations to include more models, such as GPT-4o, Claude 3.5, and Video-ChatGPT, and have updated the leaderboard accordingly. The results show that GPT-4o achieves an average score of 62.54%, Claude 54.54%, and Video-ChatGPT 33.27%:
>
> | Model           | Art & Sports | Business | Science | Health & Medicine | Embodied Tasks | Tech & Engineering | Game  | Average |
> |-----------------|--------------|----------|---------|-------------------|----------------|--------------------|-------|---------|
> | GPT-4o      | 47.87        | 91.14    | 73.78   | 83.33             | 62.94          | 75.53              | 80.32 | 62.54   |
> | Claude 3.5  | 54.58        | 63.87    | 59.85   | 54.51             | 30.99          | 58.87              | 59.44 | 54.54   |
> | VideoChatGPT| 26.84        | 39.16    | 36.45   | 53.12             | 36.60          | 41.49              | 36.55 | 33.27   |
>
> We will continue updating our benchmark by adding new models in the future.
>
>
> >- The current method still treats videos as key frame collections for solving problems. Is it possible to examine some video-specific information, such as instance consistency issues and more temporal-related problems?
>
>
> We want to clarify that the main part of MMWorld for evaluating models are using the human-annotated data and the main results in the paper are from human-annotated data.
>
> Meanwhile, 7.6% of the questions in the MMWorld are in the "temporal understanding" type, focus on temporal understanding, which includes problems related to video-specific information such as instance consistency and other temporal-related challenges. The results, as shown in Figure 4 of our paper, indicate that models like Video-LLaVA, GPT-4V, and Gemini perform the best on these temporal-related tasks.
>
>
>
> >- It is best to devise a way to confirm that benchmark's questions need to be answered in combination with video, rather than through general knowledge stored in pretrained LLMs.
>
> In addition to our experiments on synthetic data using text-only inputs, we conducted tests on our human-annotated data with and without video inputs. The results demonstrate a significant difference in performance: across the 12 evaluated models, the average accuracy without video input was only 9.67%, while with video input, the average accuracy increased to 33.04%. This clearly indicates that the benchmark's questions require the combination of video context and cannot be accurately answered using general knowledge stored in pretrained LLMs alone.

---

> > ### Comment · Area_Chair_2SMW · 2024-08-29
> > **Reminder to response to author rebuttal**
> >
> > Dear Reviewer,
> >
> > The ddl for author and reviewer discussion is approaching. Please check the author rebuttal and leave some comments to respond to author rebuttal.
> >
> > Thanks,
> >
> > Your AC

---

### Official Review · Reviewer_XiDY · 2024-07-24
**Review of submission #891**

**Rating:** 7
**Confidence:** 4
**Correctness:** Yes
**Clarity:** Yes

**Review:**

- The novelty of the proposed benchmark compared to existing ones is not explicit. What unique capabilities does your benchmark probe compared to other benchmarks (MVBench, Perception Test, EgoSchema) ? What would a model perfectly solving your benchmark unlock ?

- It is not clear how well the benchmark tests for Intuitive physics capabilities, which is probably the most important capability to have for visual world models. Many of the questions are more related to general knowledge. For example, the sample from Figure 1 is a very bad sample to showcase in my opinion, as it seems that the video is not even required to answer the proposed questions. This might be the case for more examples in the dataset where specific knowledge is required rather than temporal understanding of what is happening in the video. Examples from Figure 2, in particular the bottom left one about ‘explanation’, are actually much more interesting.

- The benchmark focuses on video QA, but a good world model should be able to make fine-grained predictions about the world, which would translate more into dense prediction and regression tasks such as tracking with occlusion or future frames prediction.

- The paper is missing a discussion and advice on how to build next generation models, with regard to how current models perform and where they make mistakes. Also why are Gemini and GPT-4 not included in the analysis of mistakes in Figure 6 ? Line 285 mentions that GPT-4 was used for selecting questions for the analysis, so GPT-4 would perform the worse in Figure 6 ? The protocol needs clarification.

- Why not train a model to output answers (A, B, C, D) instead of doing captioning and mapping with a GPT-4 evaluator ?

- Section 4.5 discusses levels of difficulty for the different questions. But “hard” for humans does not necessarily translate into ‘hard’ for machines, e.g: questions based on specific knowledge in medicine or other disciplines where humans do not necessarily have the specific knowledge, but where the machine can if it was trained on this specific knowledge. Also this kind of question could be hard but does measure temporal understanding and world modeling capabilities.

- Line 247 mentions: “for temporal understanding, Video-LLaVA performs the best”. Maybe this is just because it was trained with higher resolution, while Gemini is more like a glorified image model, trained at 1 frame per second and limited for fine-grained temporal understanding.

- Table 3 results are organized by discipline, why such emphasis on discipline which is more related to knowledge than capabilities ? It would be more interesting in my opinion to organize by capabilities. This is actually done in Figure 4 but the results from this Figure are hard to parse. The color scale is not very clear and the diagram is a bit misleading. A table or a Figure with bars would be clearer.

**Strengths:**

- The proposed benchmark is very interesting for the video understanding community. The questions are of good quality and span a broad range of domains, from sport, art to science, medicine and engineering. The breadth of evaluated reasoning tasks is also very wide, with challenging future prediction, counterfactual thinking and temporal understanding tasks. In particular the prioritization of temporal information seems to be a key factor of differentiation with existing work.

- The data annotation pipeline is interesting and the idea of bootstrapping from existing video and language models has a lot of potential for designing better benchmarks.

- The paper proposes an extensive analysis of how current models (Including GPT-4V, Gemini Pro, and many open models) perform on the benchmark, as well as an analysis on which questions current models fail. Results are also studied by domain, by reasoning task, by difficulty, and by modality (audio or video).

**Additional Feedback:**

No

**Documentation:**

Yes

**Limitations:**

Limitations are not really discussed, what capabilities of video models are not evaluated by your benchmark ?

**Opportunities For Improvement:**

Clarify novelty, answer what will this benchmark in particular unlock ? How to build next generation models ?

**Relation To Prior Work:**

Need to be clarified.

**Summary And Contributions:**

This paper presents a new question answering (QA) benchmark for evaluating video language models (VLM) in a broad variety of disciplines and reasoning tasks. The novelty comes from automatic annotation and question answer pairs generation procedure, and the breadth of domains and reasoning tasks. Many of the recent best performing VLM, such as GPT-4V and Gemini Pro are evaluated, with an in-depth analysis of where they perform well and failure cases.

---

> ### Author Rebuttal · Authors · 2024-08-17
>
> Thank you for the thoughtful and detailed review! We’re glad you found the benchmark valuable and appreciated the quality, breadth, and depth of our work. Below, we provide pointwise responses to your review:
>
> >- The novelty of the proposed benchmark compared to existing ones is not explicit. What unique capabilities does your benchmark probe compared to other benchmarks (MVBench, Perception Test, EgoSchema) ? What would a model perfectly solving your benchmark unlock ?
>
> As discussed in the paper, MMWorld differentiates itself from existing benchmarks through its comprehensive coverage of multiple disciplines and multi-faceted reasoning abilities.
> - While MVBench focuses primarily on temporal understanding, MMWorld extends this by also evaluating other critical reasoning skills such as counterfactual thinking and domain-specific expertise;
> - Perception Test emphasizes perception tasks like memory and abstraction but operates within scenarios involving a few objects manipulated by a person, which limits context variety. In contrast, MMWorld operates in an open-domain scenario with diverse scenes;
> - EgoSchema focuses on natural human activity and behavior, but it does not cover the broad range of disciplines that MMWorld does.
>
> As shown in **Table 1** of the paper, none of these benchmarks match the multi-discipline coverage that MMWorld provides. We believe solving the unique challenges presented in MMWorld is essential and necessary towards comprehensive world modeling.
> A model that can perfectly solve MMWorld would unlock the ability to perform multifaceted, multidisciplinary reasoning and the potential to serve as a world model.
>
>
>
>
> >- It is not clear how well the benchmark tests for Intuitive physics capabilities, which is probably the most important capability to have for visual world models. Many of the questions are more related to general knowledge. For example, the sample from Figure 1 is a very bad sample to showcase in my opinion, as it seems that the video is not even required to answer the proposed questions. This might be the case for more examples in the dataset where specific knowledge is required rather than temporal understanding of what is happening in the video. Examples from Figure 2, in particular the bottom left one about ‘explanation’, are actually much more interesting.
>
>
> As highlighted in Figure 2, examples that primarily test intuitive physics capabilities, such as those focused on explanation, make up 48.0% of the dataset. We have tested the performance on the example from Figure 1 using text-only inputs versus video inputs. Across the 12 models evaluated in the paper, the results showed the following average accuracies for the four questions from Figure 1:
>
> | Question Type           | With Video (%) | Without Video (%) |
> |-------------------------|----------------|-------------------|
> | Counterfactual Thinking | 58.33          | 25.00             |
> | Domain Expertise        | 75.00          | 41.67             |
> | Explanation             | 41.67          | 16.67             |
> | Future Prediction       | 66.67          | 25.00             |
>
> Additionally, when testing the models across the entire dataset, the average accuracy across the 12 models without video input was 9.67%, while with video input, it was 33.04%. This demonstrates the necessity of video context for models to provide correct answers to questions in MMWorld. We will replace Figure 1 with a more illustrative example in the revision to better showcase the benchmark's focus on multimodal video understanding.
>
>
>
> >- The benchmark focuses on video QA, but a good world model should be able to make fine-grained predictions about the world, which would translate more into dense prediction and regression tasks such as tracking with occlusion or future frames prediction.
>
> We would like to clarify that the concept of a "world model" in our paper is used in a broader, more generalized sense. Our broader definition of a world model extends beyond the traditional "world model" concept in the RL domain to include multi-discipline and multi-faceted reasoning abilities. To model the world, one also needs to understand it first. The benchmark we propose evaluates the potential of multimodal large language models (MLLMs) to serve as world models by assessing their ability to engage in multi-faceted reasoning, such as using language to describe future states, make future predictions, and perform counterfactual thinking;
> In our benchmark, 8.8% of the questions are of the “Future Prediction” type, which ask models to predict future frames in language; In addition, we also want to emphasis that our benchmark is not a sufficient testbed for world model evaluation, and we believe overcoming the unique challenges presented in MMWorld is essential and necessary towards comprehensive world modeling.

---

> ### Author Rebuttal · Authors · 2024-08-17
>
> >- The paper is missing a discussion and advice on how to build next generation models, with regard to how current models perform and where they make mistakes. Also why are Gemini and GPT-4 not included in the analysis of mistakes in Figure 6 ? Line 285 mentions that GPT-4 was used for selecting questions for the analysis, so GPT-4 would perform the worse in Figure 6 ?
>
> Regarding Figure 6, we would like to clarify that GPT-4 was used as a representative model for selecting errors that are common across MLLMs due to its strong performance. Gemini and GPT-4 were not included in the mistake analysis in Figure 6 because our focus was on evaluating the performance of current open-sourced MLLMs to provide insights and guidance for developing next-generation models. One key observation is that hallucination and lack of domain-specific knowledge were frequent errors, suggesting that accurate, well-annotated, and noise-free data is crucial for building next generation models. Additionally, Reinforcement learning from human feedback could be important for reducing hallucinations in future models. While MLLMs already possess multi-discipline world knowledge, they could still benefit from enhanced domain-specific knowledge, possibly through retrieval-based methods. We will expand on this analysis and provide more detailed suggestions in the revised paper. Additionally, we will also include the analysis of Gemini and GPT-4 in the revision.
>
> >- Why not train a model to output answers (A, B, C, D) instead of doing captioning and mapping with a GPT-4 evaluator ?
>
> We would like to clarify that the quality of the GPT-4 evaluator has been rigorously verified through human evaluation in Table 9 of our Appendix, with an error rate of only 4.76%, as shown in Table 9 of the appendix. We use GPT-4 as the evaluator to ensure that our benchmark is both solid and reliable.
>
> Thanks for the suggestion! Training a model to output the answer is a cost-effective and efficient approach compared with using GPT-4. We have now experimented with fine-tuning a Video-LLaVA model on our dataset to directly generate these outputs based on the given video and question. The model successfully produced A, B, C, D outputs for 96.28% of the questions, though it still doesn't cover the entire dataset. This indicates that while the direct output approach is promising, the use of GPT-4 as an evaluator remains the most comprehensive and fair method for ensuring consistency across all questions.
>
> We have now also tried to directly using the open-sourced Video-LLaVA model as the evaluator and perform evaluation, the performance (%) are:
> | Model             | Performance (%) |
> |-------------------|-----------------|
> | Video-Chat        | 41.96           |
> | ChatUnivi         | 39.81           |
> | mPLUG-Owl         | 38.01           |
> | PandaGPT          | 31.66           |
> | ImageBind         | 31.65           |
> | X-Instruct-BLIP   | 22.02           |
> | LWM-1M-JAX        | 16.81           |
> | Otter             | 12.08           |
> | Video-LLaMA       | 10.84           |
>
> The rankings are exactly the same as those obtained using GPT-4 in the main paper. We will release the tool that utilizes those models for evaluation, providing a cost-effective alternative to calling the GPT-4 API.
>
> >- Section 4.5 discusses levels of difficulty for the different questions. But “hard” for humans does not necessarily translate into ‘hard’ for machines, e.g: questions based on specific knowledge in medicine or other disciplines where humans do not necessarily have the specific knowledge, but where the machine can if it was trained on this specific knowledge. Also this kind of question could be hard but does measure temporal understanding and world modeling capabilities.
>
> We would like to clarify that the difficulty levels in our benchmark are defined based on the average human performance, not machine performance. As discussed in Section 4.5, we have analyzed model performance on questions of varying difficulty levels and noted that the correlation between human-defined difficulty and machine performance is not always direct. We will make these clearer in the revision.
>
> >- Line 247 mentions: “for temporal understanding, Video-LLaVA performs the best”. Maybe this is just because it was trained with higher resolution, while Gemini is more like a glorified image model, trained at 1 frame per second and limited for fine-grained temporal understanding.
>
> We agree with your observation “it was trained with higher resolution”, we will include this point and add relative discussions in our revision.
>
>
> >- Table 3 results are organized by discipline, why such emphasis on discipline which is more related to knowledge than capabilities ? It would be more interesting in my opinion to organize by capabilities. This is actually done in Figure 4 but the results from this Figure are hard to parse. The color scale is not very clear and the diagram is a bit misleading.
>
> We want to emphasis that the multi-discipline nature of the multimodal world necessitates a grasp of diverse fundamental principles and knowledges, from physics and chemistry to engineering and business, therefore, evaluating the MLLM’s capabilities regarding different disciplines is a very important aspect. Nevertheless, we will replace Figure 4 with a table in our revision to make it more clear. Additionally, we will include bar charts that organize the results by capabilities across different disciplines, providing a more comprehensive view of the evaluated models' strengths and weaknesses.
>
>
> >- Limitations are not really discussed, what capabilities of video models are not evaluated by your benchmark ?
>
>  We would like to acknowledge that is MMWorld is not a sufficient but an essential and necessary testbed for world model evaluation. We focus on multi-faceted reasoning and multi-discipline coverage. The benchmark does not systematically evaluate visual generation abilities of video models.

---

> > ### Comment · Area_Chair_2SMW · 2024-08-29
> > **Response to author rebuttal**
> >
> > Dear Reviewer,
> >
> > The ddl for author and reviewer discussion is approaching. Please check the author rebuttal and leave some comments to respond to author rebuttal.
> >
> > Thanks,
> >
> > Your AC

---

### Official Review · Reviewer_ipky · 2024-07-25

**Rating:** 8
**Confidence:** 5
**Clarity:** Yes, it's well written.

**Review:**

The paper is well-written and structured, providing a clear explanation of the MMWorld benchmark, its dataset, and the evaluation metrics used. The experimental setup is thorough, and the results are presented with detailed analysis.

The manuscript is clear and easy to follow. The figures and tables are well-designed, aiding in the understanding of the benchmark’s scope and the performance of various MLLMs.

The introduction of a benchmark like MMWorld is novel, especially in its comprehensive coverage of multiple disciplines and reasoning facets in video understanding. This sets it apart from previous benchmarks focused on narrower domains or fewer reasoning skills.

**Strengths:**

Comprehensive Benchmark: MMWorld covers a wide range of disciplines and reasoning skills, providing a robust evaluation framework.

Rich Dataset: The combination of human-annotated and synthetic datasets ensures thorough testing of MLLMs across different scenarios.

Detailed Evaluation: The paper offers in-depth analysis of MLLMs’ performance, identifying specific strengths and weaknesses.

Insightful Findings: The comparison between MLLMs and human performance provides valuable insights into current model capabilities and limitations.

**Additional Feedback:**

N/A

**Correctness:**

The claims made in the submission are well-supported by the data and experiments presented. The evaluations are conducted rigorously, and the results are interpreted accurately. The synthetic dataset is constructed soundly, ensuring a fair assessment of MLLMs’ capabilities.

**Documentation:**

Yes, it's sufficient.

**Limitations:**

The authors acknowledge the limitations of their work, including the current performance gap between MLLMs and humans, and the challenges in multi-faceted reasoning. They provide constructive suggestions for future research, emphasizing the need for more advanced models and better evaluation techniques.

**Opportunities For Improvement:**

Continous Integration with New Models:
One important aspect of benchmarks is the continuous integration of new models and the ability to facilitate comparisons. It is recommended to integrate this into benchmarks toolkits like lmms-eval for ease of use with the community. Additionally, creating a leaderboard to compare the latest results is also recommended.

**Relation To Prior Work:**

Yes, it's clearly discussed.

**Summary And Contributions:**

The paper introduces MMWorld, a benchmark for evaluating the capabilities of Multimodal Large Language Models (MLLMs) in video understanding across multiple disciplines and reasoning facets. MMWorld includes a dataset of 1,910 videos covering seven broad disciplines and 69 subdisciplines, with 6,627 question-answer pairs and associated captions. The benchmark assesses various reasoning skills such as explanation, counterfactual thinking, future prediction, and domain expertise. Evaluations on 12 MLLMs reveal that current models, including GPT-4V and Gemini, struggle with the benchmark, highlighting significant room for improvement.

---

> ### Author Rebuttal · Authors · 2024-08-17
>
> Thank you for recognizing our work and for the thoughtful and detailed review! We appreciate your positive feedback on the clarity, structure, and contributions of our paper. Here we give responses regarding your suggestions:
>
> >- Continous Integration with New Models: One important aspect of benchmarks is the continuous integration of new models and the ability to facilitate comparisons. It is recommended to integrate this into benchmarks toolkits like lmms-eval for ease of use with the community. Additionally, creating a leaderboard to compare the latest results is also recommended.
>
>
> Many thanks for your valuable suggestion! We are committed to continuously integrating new models and adding new data samples into the benchmark. The MMWorld benchmark is actively maintained and hosted at mmworld-bench.github.io. We have also established a leaderboard to facilitate comparisons of the latest results and have set up an evaluation server on https://eval.ai/web/challenges/challenge-page/2359/overview where models can be submitted for evaluation. Additionally, we also plan to integrate the benchmark into popular toolkits like lmms-eval, which will further enhance its accessibility and usability for the community.

---

> > ### Comment · Area_Chair_2SMW · 2024-08-29
> > **Response to author rebuttal**
> >
> > Dear Reviewer,
> >
> > The ddl for author and reviewer discussion is approaching. Please check the author rebuttal and leave some comments to respond to author rebuttal.
> >
> > Thanks,
> >
> > Your AC

---

### Decision · Program_Chairs · 2024-09-26

**Decision:**

Reject

**Comment:**

This paper receives mixed reviews: three accepts and one reject. While  the reviewers appreciate that this paper provides a new benchmark for video understanding involving multi-discipline multiple facts, the reviewers raised major issues. One concern raised by Reviewer 7qvY was the used terminology ("world model"). Reviewers XiDY and 7qvY also  questioned if the benchmark evaluates temporal understanding since most questions can be answered without video data. The AC agrees with Reviewer 7qvY that the term "world model" has a clear definition in the traditional RL community. World model means the ability of predicting the next world state given the current state and the possible action type. The term "world model" might thus lead to a miss-understanding for the readers. While the term "world model" can be changed, the limitations of the benchmark in terms temporal understanding have not been resolved. In fact, there are several critical issues with the proposed benchmark:

a. The proposed benchmark is composed of two subsets: manual annotated subset (3.1) and synthetic subset (3.2). There is no clear motivation why we need two kinds of evaluation subsets. It seems this work tries to put together two pieces of works. Meanwhile, the paper claims to collect 1910 videos in total for evaluation and it is not clear how many videos are human annotated and how many videos are synthetic.

b. To build a high-quality evaluation benchmark, in the AC's opinion, all these test samples should be human annotated to ensure its correctness and less biased by the annotation models. So, it is very important to detail the human annotation process in the paper. However, the description on the question annotation is very short in the paper (line 144-155, it is simply about the definition of question type, which appeared somewhere else of the paper). It is almost impossible to find any details of how annotators have been guided to design proper questions, how to generate the correct answer, and how to control the annotation process. In fact, these details are very important to ensure a high-quality benchmark and also for readers to understand the annotation process.

So based on the above reasons, the AC makes a reject recommendation to this paper.